# Role of aerosol–cloud–radiation interactions in modulating summertime quasi-biweekly rainfall intensity over South China

Hongli Chen[1], Pang-Chi Hsu[1], Anbao Zhu[1,2], and Xiaoyan Ma[3]

[1]State Key Laboratory of Climate System Prediction and Risk Management/Key Laboratory of Meteorological Disaster, Ministry of Education/Collaborative Innovation Center on Forecast and Evaluation of Meteorological Disasters, Nanjing University of Information Science and Technology, Nanjing, China

[2]Joint International Research Laboratory of Atmospheric and Earth System Sciences, School of Atmospheric Sciences, Nanjing University, Nanjing, China

[3]Key Laboratory for Aerosol-Cloud-Precipitation of China Meteorological Administration, School of Atmospheric Physics, Nanjing University of Information Science and Technology, Nanjing, China

*Correspondence to:* Pang-Chi Hsu (pangchi@nuist.edu.cn)

**Abstract.** Persistent heavy rainfall events over densely populated South China are closely linked to the intensification of quasi-biweekly (8–30-day) oscillations. This study examines whether and how aerosols influence quasi-biweekly oscillations using observational analyses and model experiments. Statistical analysis reveals a significant phase-leading relationship between increased aerosol loadings, quantified by aerosol optical depth, and subsequent enhancement of 8–30-day rainfall anomalies. At the 8–30-day timescale, aerosols primarily influence rainfall intensity through cloud microphysical processes, with radiative effects playing a secondary role. Approximately four days before enhanced rainfall events, positive aerosol anomalies contribute to increased low-level cloud water content, leading to condensational latent heat release. This low-level latent heating strengthens low-level moisture convergence and ascending motion, which uplifts cloud droplets above the freezing level. Subsequently, additional latent heat release from mixed-phase processes (freezing/deposition) further intensifies vertical motion, amplifying precipitation anomalies. Once deep convection develops, clouds absorb longwave radiation, sustaining precipitation intensification. Sensitivity experiments using the Weather Research and Forecasting model coupled with Chemistry (WRF-Chem) confirm these mechanisms, demonstrating that anthropogenic aerosol enhancement intensifies precipitation anomalies through both aerosol-cloud microphysical interactions and longwave cloud-radiative effects, with the former being more dominant. Quantitatively, aerosol-induced latent heating exceeds aerosol-induced longwave radiative heating by a factor of ~4–7 in both observations and model simulations. These findings highlight the need to improve aerosol-cloud microphysical parameterizations in operational models to enhance the accuracy of extended-range heavy rainfall predictions in South China.

## 1 Introduction

Aerosols influence clouds and precipitation through two primary mechanisms: one involves directly modifying radiation, while the other acts through their role as cloud condensation nuclei (CCN) or ice nuclei (IN) (e.g., Koren et al., 2004; Tao et al., 2012; Li et al., 2016, 2019; Zhu et al., 2022; Zhao et al., 2023; Stier et al., 2024). These are referred to as the radiative effect and the microphysical effect, respectively. The radiative effect involves the scattering and absorption of solar radiation by aerosols (i.e., the so-called "direct effect"), which commonly leads to atmospheric heating, surface cooling, stabilization of atmospheric stratification, and suppression of precipitation (Bollasina et al., 2011), but can also enhance local or remote precipitation under favorable conditions (e.g., Fan et al., 2015; Zhu et al., 2022; Wei et al., 2023). In particular, absorbing aerosols within clouds enhance cloud evaporation, thereby inhibiting cloud and precipitation formation, a phenomenon referred to as the semi-direct effect (Ackerman et al., 2000). The microphysical effect occurs when aerosols serve as CCN or IN (referred to as the "indirect effect"), increasing cloud droplet number concentration, enhancing cloud albedo (Twomey, 1977), reducing precipitation efficiency, and prolonging cloud lifetime (Albrecht, 1989). Additionally, the aerosols can invigorate deep convective cloud through freezing-induced intensification (Rosenfeld et al., 2008) and enhanced condensational heating (Fan et al., 2018), the so-called invigoration effect (Fan et al., 2025), though its significance remains debated qualitatively and quantitatively.

A recent review by Zhao et al. (2024) summarized that aerosol effects on precipitation are complex and highly variable, depending on factors such as aerosol type and concentration, precipitation characteristics and meteorological conditions, leading to substantial regional disparities. Diversity in the aerosol–precipitation relationship arises across regions and timescales due to variations in dynamic and thermodynamic conditions, aerosol properties, and multi-scale interactions. While the impacts of aerosols on short-term weather phenomena (i.e., diurnal cycles and synoptic variability of rainfall) and long-term climate variations (i.e., seasonal and interannual-decadal changes in precipitation) have been well documented (e.g., Li et al., 2016; Chen et al., 2018; Sun and Zhao, 2021; Zhao et al., 2024; Chen et al., 2025), their influences on intraseasonal rainfall variability—closely linked to persistent extreme precipitation events in the Asian monsoon regions (Hsu et al., 2016; Chen et al., 2024; Xie et al., 2024) —remain poorly understood.

Prior research has primarily focused on the effects of aerosols on the intraseasonal evolution of the Indian summer monsoon; however, no consensus has yet been reached. Some studies suggest that aerosols suppress Indian monsoon rainfall (Bhattacharya et al., 2017; Dave et al., 2017; Arya et al., 2021; Surendran et al., 2022; Debnath et al., 2023), whereas others highlight their positive contribution to enhanced monsoon precipitation variations (Manoj et al., 2011; Hazra et al., 2013; Vinoj et al., 2014; Singh et al., 2019). These contrasting findings may stem from differences in datasets and models or, more

fundamentally, from uncertainties in subseasonal aerosol–cloud–precipitation interactions (Li et al., 2016; Wang et al., 2022). In particular, the competing effects of radiative and microphysical processes may vary spatially and temporally, depending on aerosol type and emission sources (local vs. remote). Among these effects, the positive contribution of aerosol indirect effects on intraseasonal oscillations of the Indian monsoon has been emphasized (Hazra et al., 2013), mainly through the enhancement of cold-rain processes involving increased ice hydrometeors and strengthened high-level latent heating.

Influenced by active intraseasonal oscillations, persistent heavy precipitation frequently strikes densely populated southeastern China (Hsu et al., 2016), posing increasingly severe threats to socioeconomic development and the livelihoods of billions. Research on aerosol–precipitation interactions over South China in summer has predominantly examined diurnal precipitation shifts (Guo et al., 2016; Lee et al., 2016; Sun and Zhao, 2021), mesoscale rainfall intensity (Zhang et al., 2020; Xiao et al., 2023a), synoptic-scale rainfall variability (Liu et al., 2020; Guo et al., 2022), and seasonal-to-climatological rainfall changes (Wang et al., 2011; Yang and Li, 2014; Zhu et al., 2023). However, despite the importance of intraseasonal oscillations in regulating regional rainfall, few studies have examined aerosol impacts on intraseasonal variability of rainfall intensity. Thus, this study explores the role of aerosols in modulating rainfall anomalies at the quasi-biweekly (8–30-day) timescale, which is a critical factor in persistent heavy rainfall in this region (Chen et al., 2024). The relative contributions of cloud microphysical and radiative processes to rainfall intensification are examined through observational analysis and model experiments. The remainder of the paper is structured as follows. Section 2 describes the datasets and methodologies used in this study. Section 3 analyzes the observed aerosol impacts on quasi-biweekly precipitation anomalies. Section 4 presents the experimental results from the Weather Research and Forecasting model coupled with Chemistry (WRF-Chem). Concluding remarks and discussion are provided in Section 5.

## 2 Data and methods

### 2.1 Datasets

Daily precipitation data were obtained from the National Oceanic and Atmospheric Administration (NOAA) Climate Prediction Center (CPC) dataset (Chen et al., 2008) with a horizontal resolution of 0.5°. Deep convection activity was analyzed using daily interpolated outgoing longwave radiation (OLR) data from NOAA at a 2.5° grid resolution (Liebmann and Smith, 1996).

To investigate the effects of aerosols on radiative and moisture processes, daily mean aerosol optical depth (AOD), radiative fluxes, specific humidity and circulation variables from the Modern-Era Retrospective Analysis for Research and Applications version 2 (MERRA-2), produced by the National Aeronautics and Space Administration (NASA) Global Modeling and Assimilation Office (Gelaro et al.,

2017), were collected at a spatial resolution of 0.5° × 0.625° (latitude × longitude) on 42 vertical levels. MERRA-2 provides the complete set of variables required for atmospheric radiation and moisture budget quantifications, whereas other reanalyses and observations lack some of these key variables. Even so, to ensure that our conclusions are not dataset-dependent, we compared budget analysis results with those derived from the fifth-generation European Center for Medium-Range Weather Forecasts (ECMWF) atmospheric reanalysis (ERA5; Hersbach et al., 2020), which has a spatial resolution of 0.25° × 0.25° and 37 vertical levels. To further reduce uncertainties inherent in reanalyses, we also employed radiative fluxes from Clouds and the Earth's Radiant Energy System Synoptic products (CERES-SYN; Rutan et al., 2015) at 1° resolution, and AOD from the Moderate Resolution Imaging Spectroradiometer (MODIS) Collection 6 Level-3 aerosol product onboard the Terra satellite (Levy et al., 2013) at 1° resolution.

To examine aerosol influences on cloud microphysical processes, daily two-dimensional cloud parameters, including liquid cloud fraction, ice cloud fraction, cloud top pressure, cloud droplet effective radius and ice water path, were obtained from the MODIS Collection 6 Level-3 cloud product (Platnick et al., 2017). These cloud properties have been widely used to analyze aerosol indirect effects (e.g., Zhou et al., 2020; Jia et al., 2024). Although CloudSat provides three-dimensional cloud products (Austin et al., 2009), substantial temporal gaps prevent its use for analyzing continuous sequences of aerosol–cloud–precipitation interactions at intraseasonal timescales. Thus, three-dimensional liquid and ice cloud water contents were instead taken from MERRA-2 and ERA5 to evaluate vertical cloud structures.

All datasets cover the period from 2000 to 2021, with analyses focused on the boreal summer season (May–September, MJJAS).

**2.2 Diagnostic method**

To investigate the processes regulating precipitation anomalies, we diagnosed the budget equation of column-integrated moisture perturbation:

$$\langle \frac{\partial q}{\partial t} \rangle' = -\langle \mathbf{V} \cdot \nabla q \rangle' - \langle q \nabla \cdot \mathbf{V} \rangle' - \langle \frac{\partial}{\partial p}(\omega q) \rangle' - \langle \frac{Q_2}{L} \rangle', \tag{1}$$

where the primes denote the 8–30-day component derived using the Lanczos filtering method (Duchon, 1979), angle brackets indicate a mass-weighted vertical integration from 1000 to 100 hPa level, $\mathbf{V}$ is the horizontal wind field, $q$ is the specific humidity, $\omega$ is vertical pressure velocity, $\nabla$ is the horizontal gradient operator, $p$ is pressure, $L$ is the latent heat of condensation, and $Q_2$ represents the latent heating due to condensation/evaporation processes and subgrid-scale moisture flux convergences (Yanai et al., 1973). The terms on the right-hand side of Eq. (1) correspond to the horizontal moisture advection, moisture

convergence, vertical moisture flux, and the moisture source or sink associated with phase transitions (i.e., precipitation and evaporation) at the 8–30-day timescale.

The radiative effect is quantified using MERRA-2 data. The difference between net surface radiation and net top-of-atmosphere (TOA) radiation is defined as the atmospheric net radiative change, which includes both longwave and shortwave components. Longwave and shortwave radiation can be further categorized into clear-sky radiation and cloud radiative effects. Following Lin and Chen (2023), the radiative effects driven by aerosols and greenhouse gases are isolated linearly under assumed clear-sky and/or no-aerosol conditions:

$$\begin{cases} Cld\_RE = Rad - Rad_{clr} \\ GHG\_RE = Rad_{clr+no\_aer} \\ Aer\_RE = Rad_{clr} - Rad_{clr+no\_aer} \end{cases}, \qquad (2)$$

where $Rad$ represents the atmospheric net radiation flux, $Cld\_RE$ is the cloud radiative effect, $Aer\_RE$ is the aerosol radiative effect, and $GHG\_RE$ is the greenhouse gas radiative effect. The subscripts "$clr$" and "$clr+no\_aer$" denote radiation fluxes under clear sky conditions and clear-sky/aerosol-free conditions, respectively. A positive value indicates a downward flux, contributing to atmospheric warming. Note that the $Aer\_RE$ in Eq. (2) captures only the direct radiative effect, as this method does not account for indirect aerosol effects on cloud properties and droplets formation (Lin and Chen, 2023).

The bootstrap method (Mudelsee et al., 2014) is applied to assess the statistical significance of results from observations and model experiments. The procedure consists of the following steps: first, paired resampled datasets are generated through random sampling with replacement from both experimental groups. Next, target statistical metrics are computed for each resampled pair. Finally, this process is repeated 1,000 times, and the 90% confidence interval is determined by selecting the 5th and 95th percentile values.

**2.3 WRF-Chem experiments**

The covariation of aerosols and meteorological conditions complicates the attribution of causality between aerosol–cloud–radiation interactions and precipitation in southern China through observational analyses. To address this, we conducted a series of experiments using the WRF-Chem version 4.2.2 (Grell et al., 2005; Fast et al., 2006) to support the observed mechanisms responsible for aerosol impacts on clouds and precipitation, although uncertainties remain due to the dependence on emission inventories, physical parameterizations, and initial and boundary conditions. To increase computational efficiency, we adopted a single-domain configuration, consistent with previous studies focusing on aerosol–cloud–radiation feedbacks beyond weather timescales (e.g., Zhang et al., 2010; Chen et al., 2018). The model

domain covers most of China (approximately 8°–43°N, 89°–140°E), with a horizontal resolution of 20 km and 40 vertical levels extending from the surface to 50 hPa.

The meteorological initial and boundary conditions are obtained from the National Centers for Environmental Prediction Final Analysis (NCEP-FNL) dataset, which is provided on a 1° × 1° grid and updated every six hours. Sea surface temperature (SST) data are sourced from the NCEP real-time global SST analysis. Simulations using NCEP-FNL data as WRF-Chem input better capture the AOD evolution compared to those driven by higher-resolution ERA5 data (figure not shown). To better reproduce the observed circulation and aerosol pattern, grid analysis nudging is applied only during the spin-up period (Abida et al., 2022), allowing meteorological fields to freely interact with aerosols during the analysis period. While nudging could potentially constrain aerosol feedbacks (He et al., 2017), our sensitivity tests confirm that it does not affect the main conclusions (figure not shown).

The primary parameterization schemes used in the simulations are listed in Table 1, following the model configurations of Zhang et al. (2021). To comprehensively investigate aerosol indirect effects, the Morrison double-moment microphysics scheme (Morrison et al., 2009) and the Grell–Freitas cumulus scheme (Grell and Freitas, 2014) are employed. Shortwave and longwave radiation are parameterized using the Rapid Radiative Transfer Model for General Circulation Models (RRTMG) (Iacono et al., 2008) with aerosol radiative effects included, as the Morrison scheme enables direct communication of cloud droplet, ice, and snow effective radii to the radiation module.

Key chemistry options include the Carbon-Bond Mechanism version Z (CBMZ) scheme (Zaveri and Peters, 1999) for gas-phase chemistry, the Model for Simulating Aerosol Interactions and Chemistry (MOSAIC) 4-bin scheme (Zaveri et al., 2008) for aerosols, and the Fast-J scheme (Fast et al., 2006) for photolysis. Anthropogenic emissions are derived from the Multi-resolution Emission Inventory for China (Li et al., 2017a) and the MIX inventory (Li et al., 2017b) for regions outside China. Biogenic emissions are calculated online using the Model of Emissions of Gases and Aerosols from Nature (MEGAN) (Guenther et al., 2012). Dust emissions are simulated using the original Goddard Chemistry Aerosol Radiation and Transport (GOCART) dust emission scheme (Ginoux et al., 2001). Although the GOCART scheme may underestimate dust aerosol concentrations in Asia (Zhao et al., 2020), no emission scaling or tuning was applied in this study because our simulations focus on anthropogenic aerosols and summer rainfall in South China, where dust emissions contribute minimally. Biomass burning emissions are obtained from the high-resolution fire emissions dataset based on the Fire Inventory from National Center for Atmospheric Research (NCAR) version 1.5 (Wiedinmyer et al., 2011).

Table 1. Main parameterization schemes used in the WRF-Chem simulations

| Option name | Schemes | References |
| --- | --- | --- |

| | | |
|---|---|---|
| Microphysics | Morrison double-moment | Morrison et al. (2009) |
| Cumulus | Grell–Freitas scheme | Grell and Freitas (2014) |
| Longwave radiation | RRTMG | Iacono et al. (2008) |
| Shortwave radiation | RRTMG | Iacono et al. (2008) |
| Surface layer | Monin–Obukhov | Pahlow et al. (2001) |
| Land surface | Unified Noah | Chen et al. (2010) |
| Boundary layer | Yonsei University | Hong et al. (2006) |
| Aerosol chemistry | MOSAIC | Zaveri et al. (2008) |
| Gas chemistry | CBMZ | Zaveri and Peters (1999) |
| Photolysis | Fast-J | Fast et al. (2006) |

The case selected for simulation is a persistent heavy rainfall event over South China in July 2015, lasting over five days and characterized by high aerosol loading before the onset of heavy precipitation. To minimize uncertainties in the modeling outcomes, we perform six ensemble simulations for each experiment by perturbing the initial and boundary conditions. Specifically, the ensemble members start at one-day intervals from 4 to 9 July 2015, with all simulations ending on 6 August 2015. To allow locally emitted aerosols to become sufficiently mixed and reach a quasi-equilibrated distribution, we adopted spin-up times of 1–6 days, consistent with previous studies (e.g., Zhu et al., 2022). The first few days of each run (4–9 July) are discarded, and the analysis focuses on 10 July–6 August 2015.

**3 Observed aerosol effects on enhanced 8–30-day rainfall anomalies**

In the East Asian monsoon region, the quasi-biweekly (8–30-day) oscillation is particularly strong in southern China (18°–32°N, 108°–123°E; box in Fig. 1a) and has been identified as a key trigger of persistent heavy rainfall events (Chen et al., 2024). Meanwhile, aerosol emissions, measured by AOD, also exhibit significant quasi-biweekly variability (Fig. 1b), despite their maximum concentrations occurring in North China.

To examine whether aerosols in southern China are correlated with local extreme precipitation, we analyze the evolution of AOD (blue curve in Fig. 1c) in relation to persistent heavy rainfall events (blue bars in Fig. 1c), which are defined as daily precipitation exceeding the 90th percentile for at least three consecutive days. An increase in AOD is observed before the onset of persistent heavy rainfall, followed by a decline during the heavy rainfall period due to wet deposition. The evolution of both the heavy rainfall and AOD closely follow their respective 8–30-day components (red and purple curves in Fig. 1c),

indicating that both fields are strongly modulated by the quasi-biweekly oscillation. In contrast, neither AOD nor rainfall displays quasi-biweekly variability in composites of normal rainfall cases, where amplitudes remain around the climatological mean (45th–55th percentiles). No phase-leading relationship between AOD and rainfall is observed in these cases (Fig. 1d). The distinct evolution of AOD associated with different rainfall intensities suggests that the phase-leading increase in AOD could play a role in triggering persistent heavy rainfall events, with their connections primarily manifesting at the quasi-biweekly timescale.

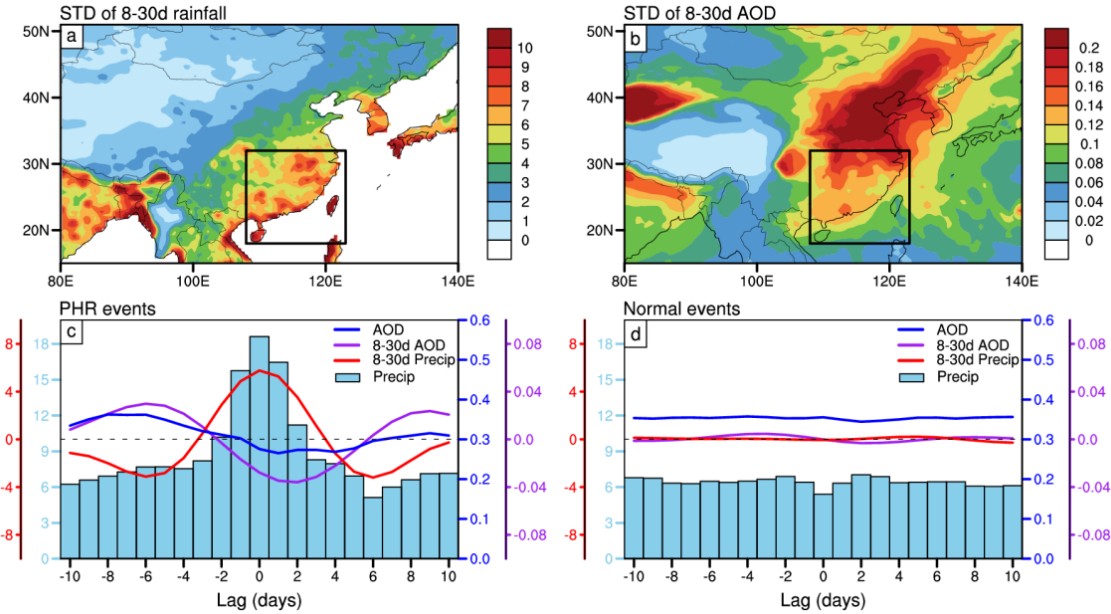

Figure 1. Spatial distributions of the standard deviation of (a) 8–30-day-filtered precipitation anomalies (mm d$^{-1}$) and (b) 8–30-day-filtered aerosol optical depth anomalies (AOD; unitless) during MJJAS from 2000 to 2021, based on CPC rainfall data and MERRA-2 AOD data, respectively. The black boxes indicate the study domain of southern China (18°–32°N, 108°–123°E). (c) Composites of daily precipitation (light blue bars, left $y$-axis in light blue, mm d$^{-1}$), 8–30-day-filtered precipitation (red curve, left $y$-axis in red, mm d$^{-1}$), AOD (blue curve, right $y$-axis in blue, unitless), and 8–30-day-filtered AOD (purple curve, right $y$-axis in purple, unitless) associated with persistent heavy rainfall events over South China. The $x$-axis is centered at "0", representing the median timing of individual rainfall events, with negative and positive values indicating periods before and after the event. (d) As in (c) but for the composites of normal-intensity rainfall events.

Focusing on the quasi-biweekly timescale, we further examine whether the leading phase of AOD is consistently present and essential for enhanced 8–30-day rainfall anomalies, based on statistical analyses (Fig. 2). If not, we explore under what conditions AOD contributes positively to rainfall intensification. Composite analysis shows that AOD exhibits positive anomalies beginning about six days before the peak of 8–30-day rainfall events (Day 0; red curve in Fig. 2a; blue curve for AOD). Among all 8–30-day rainfall events (249 cases), the majority (about 87%) are preceded by a leading phase of AOD anomalies, supporting a potential causal role of antecedent AOD anomalies in subsequent precipitation enhancement.

The consistent evolution of AOD from MODIS and ERA5 datasets (Figs. S1a–b) demonstrates the robustness of intraseasonal AOD variations associated with heavy rainfall events. Validation with long-term reconstructed gridded datasets of particulate matter smaller than 2.5 μm ($PM_{2.5}$) confirms these findings (Fig. S1c), indicating that the observed leading phase of aerosols is robust, regardless of whether AOD or $PM_{2.5}$ is used as the proxy for aerosol concentrations.

The composite analysis in Fig. 2a illustrates the overall lead-lag phase relationship between AOD and rainfall anomalies but does not directly quantify their correlation. To further assess this correlation, Fig. 2b examines how subsequent rainfall anomalies vary with the amplitude of preceding AOD anomalies during Days –6 to –1, considering all 8–30-day rainfall events. Although the correlation coefficient between the two parameters is relatively low (0.12; red line in Fig. 2b), it is statistically significant at the 90% confidence level due to the large sample size. A visual inspection suggests that this positive correlation is not well established when rainfall anomalies are weak (symbols to the left of the dashed line in Fig. 2b). However, when focusing on rainfall events with greater amplitude, the correlation becomes stronger. Specifically, for events with rainfall amplitude exceeding 0.25 standard deviation (σ) (blue line, symbols to the right of the dashed line in Fig. 2b), the correlation coefficient increases to 0.25, which is significant at the 95% confidence level. These results indicate that the effect of preceding AOD increases on subsequent rainfall enhancement is not linear.

To determine the threshold of rainfall amplitude significantly modulated by AOD magnitudes, we further compute the correlation coefficients between AOD anomalies preceding rainfall events and the amplitude of subsequent rainfall anomalies by categorizing events into different amplitude bins (*x*-axis in Fig. 2c). The results show that the correlation coefficient reaches 0.2, becoming significant at the 90% confidence level, when rainfall anomalies exceed the mean of all events (i.e., pink bar on the *x*-axis is at 0). The coefficient further increases to 0.25, surpassing the 95% confidence level, for events with rainfall amplitudes above 0.25σ. The correlation is even stronger, reaching 0.39, for heavy rainfall events with amplitudes exceeding 1σ (rightmost pink bar in Fig. 2c). These findings suggest that higher aerosol concentrations before rainfall occurrence are conducive to rainfall amplification, particularly for events with above-average amplitude. This behavior aligns with previous studies at the synoptic and decadal timescales (Wang et al., 2011; Yang et al., 2018; Su et al., 2020; Shao et al., 2022; Xiao et al., 2023b), which emphasize that the aerosols tend to suppress light rainfall while enhancing heavy convective precipitation.

The statistical results in Fig. 2 indicate that AOD anomalies have a more pronounced impact on 8–30-day precipitation events exceeding a certain intensity threshold (e.g., 0.25σ), with stronger antecedent AOD anomalies leading to amplified subsequent precipitation anomalies, although the correlation coefficients are modest (r=0.25–0.39). These modest values highlight the complexity of rainfall

intensification mechanisms, which involve circulation anomalies that may be induced by, or independent of, AOD variations. To investigate the physical processes through which AOD contributes to rainfall intensification, we compared enhanced 8–30-day rainfall cases (amplitude greater than 0.25σ) under varying AOD intensities prior to the event, categorized as High AOD–Strong Precipitation (HA–SP) and Low AOD–Strong Precipitation (LA–SP) cases. Thus, preceding AOD cases are classified as high or low based on whether the standardized AOD anomaly is positive or negative. To distinguish clean and polluted conditions and ensure balanced sample sizes, LA–SP cases with AOD anomalies between −0.4σ and 0 (approximately 40th–60th percentiles) were excluded (Fig. 2b).

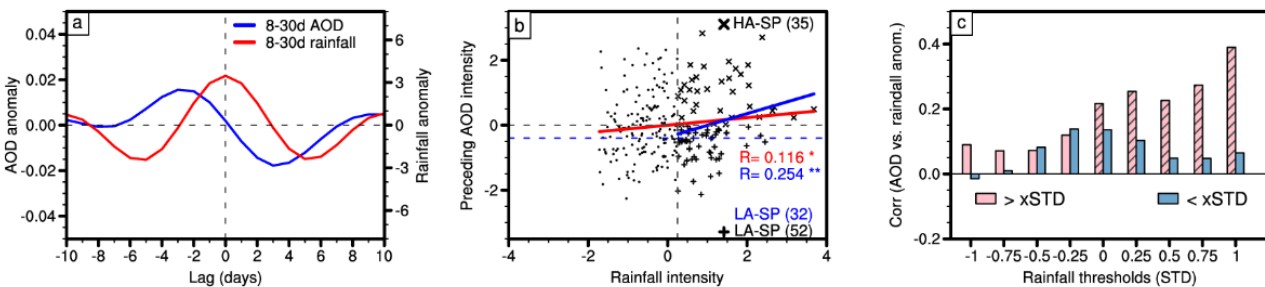

Figure 2. (a) Evolution of 8–30-day precipitation (red curve, mm d⁻¹) and 8–30-day AOD (blue curve, unitless) associated with quasi-biweekly precipitation events, defined as periods with positive 8–30-day rainfall anomalies over South China. These results are based on CPC rainfall data and MERRA-2 AOD data. Day 0 denotes the peak of rainfall events, while negative and positive values on the *x*-axis indicate days before and after the peak, respectively. (b) Scatterplot of rainfall intensity (i.e., 8–30-day-filtered rainfall anomalies at Day 0; *x*-axis, mm d⁻¹) versus preceding AOD intensity (represented by the peak value of 8–30-day-filtered AOD anomalies during Days –6 to –1; *y*-axis, unitless), with both variables normalized by their climatology. The red line represents the linear regression fitted to all cases, while the blue line corresponds to events where rainfall intensity exceeds 0.25σ. The correlation coefficients between the two variables are shown in the bottom right corner, with single and double asterisks indicating significance at the 90% and 95% confidence levels, respectively. Enhanced 8–30-day rainfall events (>0.25σ) are classified into High AOD–Strong Precipitation (HA–SP; cross sign) and Low AOD–Strong Precipitation (LA–SP; plus sign) categories, with case counts displayed in each quadrant. To maintain nearly equal sample sizes for these two groups, the LA–SP classification is determined using an AOD anomaly threshold of -0.4σ (blue dashed line and blue text). (c) Correlation coefficients between preceding AOD intensity and rainfall intensity at Day 0 for different rainfall amplitude thresholds (bins with 0.25σ intervals). Pink bars represent correlations when 8–30-day rainfall exceeds a given threshold, while blue bars show correlation coefficients for the remaining events. Hatching indicates correlations significant at the 90% confidence level.

Composite patterns of convection and circulation during the early and peak phases of HA–SP and LA–SP events, along with their differences, are shown in Fig. 3 based on MERRA-2 data, and are consistent with those derived from ERA5 (Fig. S2). At four days before the occurrence of heavy rainfall events (Day –4), both types of events exhibit suppressed convection of similar magnitude but with distinct wind anomalies (contours and vectors in Figs. 3a, d, g), which likely explain their differing AOD behaviors. HA–SP events are associated with anomalous northerlies that transport aerosols from North

China to South China, increasing regional AOD, whereas LA–SP events exhibit an opposite transport pattern. Moreover, the anomalous moisture convergence (vectors in Fig. 3g) coincides with significant AOD enhancement (shading in Fig. 3g) and intensified convection over coastal Southern China (contours in Fig. 3g), implying the aerosol effects on moistening process. Approximately two days before the heavy rainfall events, convective anomalies (indicated by negative OLR anomalies) were observed over South China in both cases (contours in Figs. 3b, e). However, in high AOD cases (HA–SP), convection and moisture convergence signals are noticeably stronger than in low AOD cases (LA–SP) (Fig. 3h). At the peak of the heavy rainfall event (Day 0), strong convection, cyclonic circulation anomalies, and significant moisture convergence dominate South China (Figs. 3c, f), coinciding with AOD reductions due to wet scavenging. Notably, convective anomalies at Day 0 are more intense in HA–SP events than in LA–SP events (Fig. 3i), suggesting that preceding positive AOD anomalies may amplify subsequent convective activity through enhanced moisture convergence.

Based on the moisture budget diagnosis, we further examine the processes responsible for increasing moisture and enhancing rainfall amplitude in HA–SP cases, comparing the budget terms with those in LA–SP events (Figs. 4a–c). In both cases, moisture convergence ($-\langle q\nabla \cdot V\rangle'$, green curves in Figs. 4a–b) demonstrated a growth of 2–3 mm $d^{-1}$ at Day 0, accounting for ~50% of the positive rainfall anomalies (red curves in Figs. 4a–b) and serving as the primary moisture source. Moreover, moisture convergence also explains the differences in rainfall amplitude between HA–SP and LA–SP events (green curve in Fig. 4c), which exhibited a significant increase of 1–1.8 mm $d^{-1}$ during Days –4 to 0. The moisture sink associated with latent heating is in phase with rainfall and offsets the moisture source from convergence with a reduction of 0.9–1.6 mm $d^{-1}$ during Days –3 to 1 (cyan curve in Fig. 4c). Nevertheless, their combined effect still contributes positively to heavy rainfall occurrence (red curve in Fig. 4c). The temporal evolution of these key terms reveals their sequential influence. An increment of ~0.1 in AOD at Day –4 slightly precedes the enhancement of moisture convergence (blue and green curves in Fig. 4c), indicating that the aerosols could play roles in moistening process, which may subsequently lead to intensification of 0.7–1 mm $d^{-1}$ in rainfall during Days –2 to 4 (red curve in Fig. 4c). At Day 0, the intensity of quasi-biweekly precipitation is increased by ~20%. The horizontal moisture advection and vertical moisture flux (magenta curves and pink curves in Figs. 4a–b) make relatively minor contributions, and their differences between HA–SP and LA–SP events are not statistically significant. Overall, the key moisture processes associated with quasi-biweekly precipitation events and their modulation by aerosols are consistently shown in the ERA5 reanalysis (Figs. S3a–c).

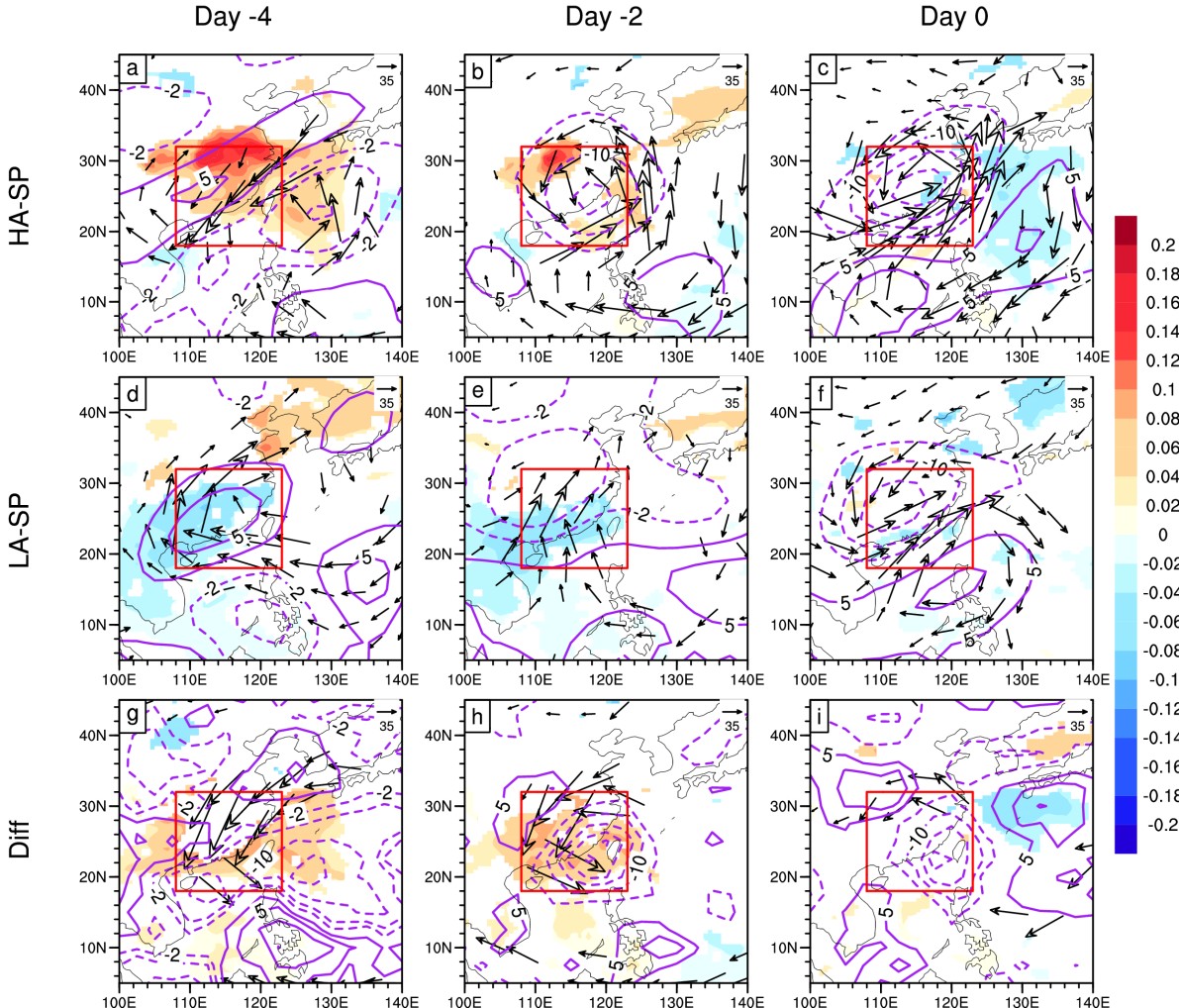

Figure 3. (a)–(c) Composite evolution of 8–30-day outgoing longwave radiation (OLR; purple contours, W m⁻²), AOD (shading, unitless), and column-integrated moisture flux (vectors, kg m⁻¹ s⁻¹) at 4, 2, and 0 days before the occurrence of HA–SP events, respectively. The results are based on NOAA OLR data, MERRA-2 AOD and meteorological data. (d)–(f) and (g)–(i) Similar to (a)–(c), but for the composite results of LA–SP events and the differences between HA–SP and LA–SP events, respectively. Only AOD and moisture flux anomalies with statistically significant changes at the 90% confidence level are shown. The red box outlines the study domain of South China.

Building on the observed lead-lag relationship between aerosols and moistening process (Figs. 4a–c), the next question is: through what physical processes do aerosols enhance moisture convergence and rainfall? Previous studies have shown that aerosols primarily influence summer precipitation in South China through cloud microphysical effects, as clouds in this region are readily invigorated by the aerosols which are less absorbing and highly hygroscopic (Fan et al., 2008; Yang et al., 2016). To investigate this mechanism, we analyzed several key cloud-related parameters for HA–SP and LA–SP events (Figs. 4d–f). Although the temporal evolution of cloud properties is similar in both cases (Figs. 4d–e), their magnitudes exhibit notable differences (Fig. 4f). At Day –4, the anomalous liquid cloud fraction increased significantly (purple curve in Fig. 4f). The most intense increase prior to heavy rainfall in HA–SP events is observed in the ice water path (orange curve in Fig. 4f), which rises by 22.2–26.8 g m⁻² during Days –

3 to −1. This magnitude of increase is comparable to the cloud water path enhancements under pollution reported in Zhou et al. (2020). These changes are likely linked to precursor moisture convergence anomalies (green curve in Fig. 4c), which enhance latent heating in the lower troposphere. The resulting low-level warming destabilizes atmospheric stratification, facilitating the uplift of abundant cloud droplets above the freezing level. Consequently, the ice water path increases markedly (orange curve and gray asterisk in Fig. 4f), whereas the ice cloud fraction shows a modest enhancement of ~0.01 at Day −3. The statistically insignificant increase in ice cloud fraction may reflect a limited reduction in the liquid cloud fraction and persistent supercooled droplets. These conditions favor interactions between supercooled droplets and cloud ice particles.

Subsequently, deep convective clouds develop, characterized by a reduction in cloud top pressure (magenta curve in Fig. 4f) and an increase in ice cloud fraction (light blue curve in Fig. 4f), alongside a gradual weakening of the liquid cloud fraction (purple curve in Fig. 4f). Additional latent heat released from freezing and deposition could further enhance upward motion, collectively amplifying the intensity of 8–30-day heavy precipitation events. Note that both cases exhibit positive anomalies in the effective radius of cloud droplets before the onset of heavy precipitation (cyan curves in Figs. 4d–e), although the difference between them is not statistically significant. This suggests that aerosol impacts on the warm-rain processes are not the primary driver of the 8–30-day rainfall enhancement, with aerosol indirect effects likely exerted through ice-phase microphysical processes.

In addition to aerosol–cloud interactions, changes in liquid and ice cloud fractions also influence atmospheric radiation variations. The right panels of Fig. 4 illustrate the longwave and shortwave radiative effects induced by clouds, aerosols, and greenhouse gases. During HA–SP events, the dominant longwave cloud-radiative effect closely follows the temporal evolution of 8–30-day precipitation anomalies (magenta curve in Fig. 4g), supporting the maintenance of heavy precipitation intensity (Chen et al., 2024). This behavior is consistent with estimates from CERES-SYN (Fig. S4). Although the amplitude is relatively small, the significant increase in GHG-induced longwave heating prior to heavy rainfall (orange curve in Fig. 4i) may be associated with the preceding phase of moistening driven by moisture convergence anomalies (green curve in Fig. 4c). Focusing on the aerosol effect, the aerosol-induced direct shortwave atmospheric heating (light blue curve in Fig. 4i) is small but in phase with AOD anomalies (blue curve in Fig. 4c). This suggests that some absorbing aerosols, such as anthropogenic black carbon, may contribute to shortwave radiation absorption over southern China. However, due to the predominance of nonabsorbing aerosols in this region (Lee et al., 2007; Huang et al., 2014; Yang et al., 2016), the resultant aerosol shortwave radiative forcing (Aer_SW) remains relatively weak (light blue curve in Fig. 4i).

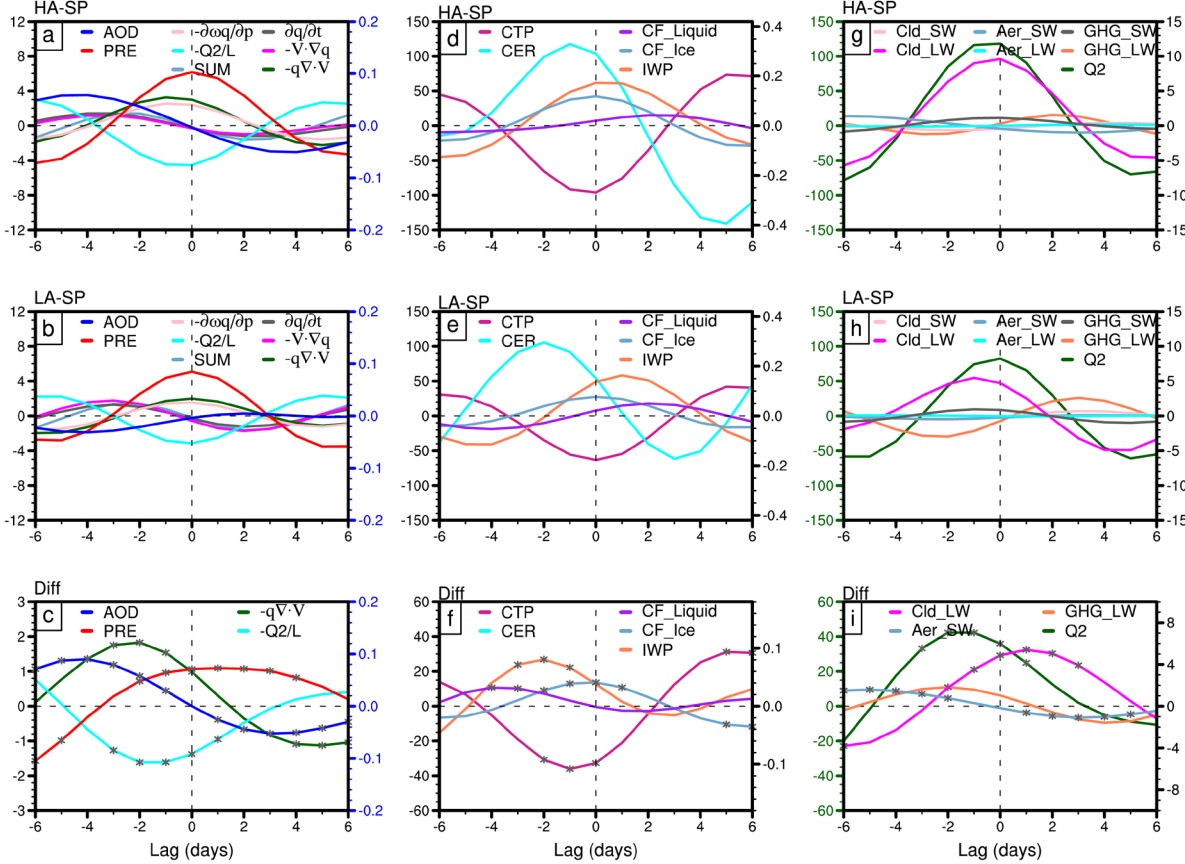

Figure 4. As in Fig. 2a, but for the composite evolution of 8–30-day (a) precipitation (red curve; left $y$-axis, mm d$^{-1}$), AOD (blue curve; right $y$-axis in blue, unitless), and individual moisture budget terms (various colored curves representing each budget term in Eq. (1); left $y$-axis, mm d$^{-1}$) associated with HA–SP events. (d, g) As in (a), except that (d) shows the 8–30-day evolution of cloud top pressure (CTP; magenta curve; left $y$-axis, hPa), ice water path (IWP; orange curve; left $y$-axis, g m$^{-2}$), cloud droplet effective radius (CER; orange curve; right $y$-axis, μm), liquid cloud fraction (CF_Liquid; purple curve; right $y$-axis, unitless), and ice cloud fraction (CF_Ice; light blue curve; right $y$-axis). Panel (g) presents 8–30-day column-integrated latent heat heating ($Q_2$; green curve; left $y$-axis in green, W m$^{-2}$) and 8–30-day radiative budget terms calculated from Eq. (2) (right $y$-axis, W m$^{-2}$), including longwave/shortwave cloud radiative effects (Cld_LW/Cld_SW; magenta curve/pink curve), longwave/shortwave aerosol direct radiative effects (Aer_LW/Aer_SW; cyan curve/light blue curve), and longwave/shortwave greenhouse gas radiative effects (GHG_LW/GHG_SW; orange curve/gray curve). (b, e, h) and (c, f, i) are similar to (a, d, g), but represent the composite results for LA–SP events and the differences between HA–SP and LA–SP events, respectively. In panels (c) and (i), only terms with statistically significant differences at the 90% confidence level are shown, with significant periods marked by gray asterisks. These results are based on MERRA-2 budget variables and MODIS cloud parameters.

Latent heating, a direct product of phase transition processes, serves as an indicator of aerosol effects on cloud microphysical properties (Zhu et al., 2024; Fan et al., 2025). Both the aerosol-induced direct radiative effects (light blue curve in Fig. 4i) and longwave cloud-radiative effects (magenta curve in Fig. 4i), with magnitudes of approximately 2–5.5 W m$^{-2}$, are significantly smaller than atmospheric latent heating associated with moisture processes, which exceeds 40 W m$^{-2}$ (green curve in Fig. 4i). Quantitatively, aerosol-induced latent heating is approximately seven times greater than aerosol-induced

longwave radiative heating. The relative contributions and quantitative ratios between the two variables are similarly shown in the ERA5 data (Figs. S3d–f). This indicates that aerosol–cloud microphysical effect plays a dominant role in enhancing heavy precipitation at the 8–30-day timescale, as demonstrated by the diagnostic frameworks of moisture and radiative budget analyses. The latent heating magnitude (~40 W m⁻²) is also comparable to values reported in previous studies of tropical intraseasonal convection (~60 W m⁻²; Fu et al., 2020) and monsoonal extreme precipitation (~50 W m⁻²; Chyi et al., 2023).

To confirm the stepwise contribution of aerosol-induced low-level moistening to high-level freezing and deposition, ultimately leading to deep convection and rainfall intensification, as discussed in Fig. 4, we further analyze the time-height evolution of latent heating, vertical velocity, and liquid/ice cloud water content differences between HA–SP and LA–SP events (Fig. 5). As shown in Fig. 5a, low-level liquid cloud water content increases below 500 hPa, coinciding with peak AOD at four days (Day –4) before the occurrence of heavy rainfall events (Fig. 4). The condensation process in shallow convection releases latent heat, forming positive latent heating anomalies at 850–700 hPa (Fig. 5c), which enhances low-level moisture convergence and upward motion (Fig. 5d). This process provides favorable conditions for the uplift of cloud droplets above the freezing level. From Days –3 to –1, strong latent heating (Fig. 5c) and ascending motion anomalies (Fig. 5d) develop above 500 hPa, corresponding to the increased ice cloud water content in the middle and upper troposphere (Fig. 5b). The enhanced conversion of supercooled cloud water to ice hydrometeors through freezing and deposition further releases latent heat, reinforcing vertical motion and convection. These processes ultimately intensify rainfall anomalies, a finding that is also confirmed by ERA5 (Fig. S5).

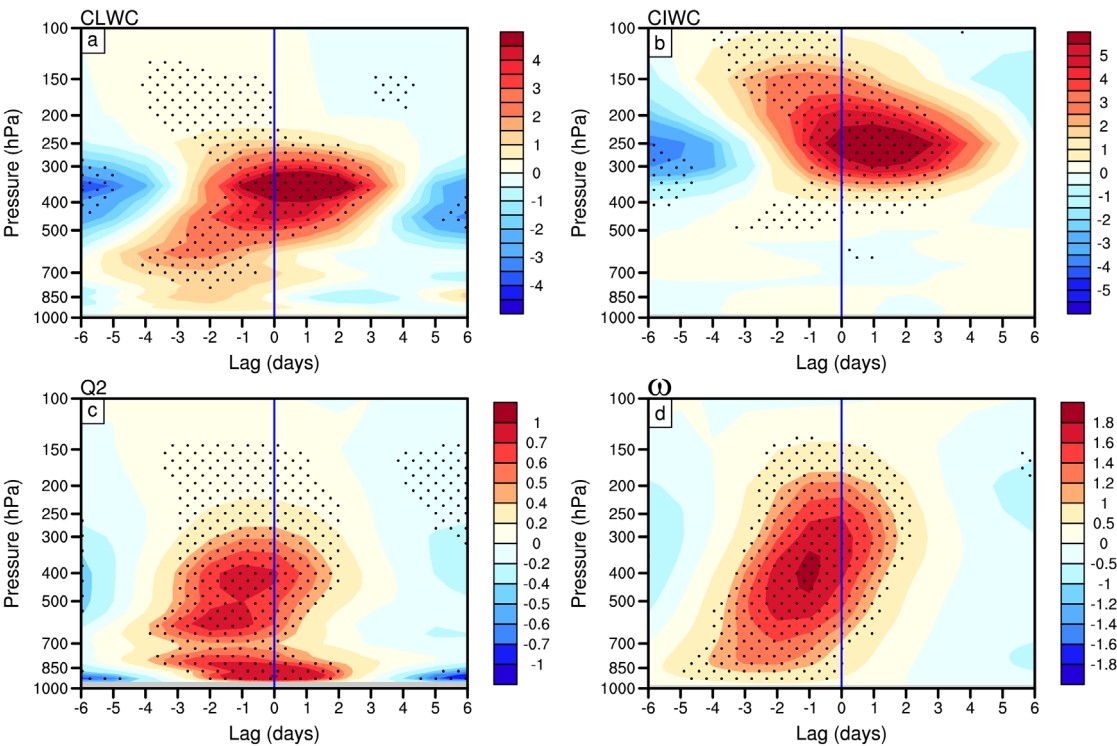

Figure 5. Height–time evolution of 8–30-day anomalies in (a) liquid cloud water content (CLWC; $10^{-6}$ kg kg$^{-1}$), (b) ice cloud water content (CIWC; $10^{-6}$ kg kg$^{-1}$), (c) latent heat heating (K d$^{-1}$), and (d) vertical velocity anomalies ($-10^{-2}$ Pa s$^{-1}$) between the HA–SP and LA–SP events, based on the MERRA-2 data. Stippling denotes differences statistically significant at the 90% confidence level. Vertical blue lines denote the peak timing of heavy rainfall event (Day 0).

## 4 Support for mechanisms using WRF-Chem experiments

Due to the complex interactions between aerosols and rainfall, as well as the limited availability of observational data on cloud microphysical properties, we employ the WRF-Chem model to verify the mechanisms identified in Sect. 3. Following the methodology of Guo et al. (2022) and Yun et al. (2024), we conduct two sets of simulations to assess the effects of anthropogenic aerosols. In the control experiment (CTRL), anthropogenic aerosol emissions are kept unchanged, while in the sensitivity experiment (CLEAN), emissions are scaled to 10% of those in the CTRL run. Hence, we examine differences in cloud microphysics, radiation, and surface rain rates between the CTRL and CLEAN simulations.

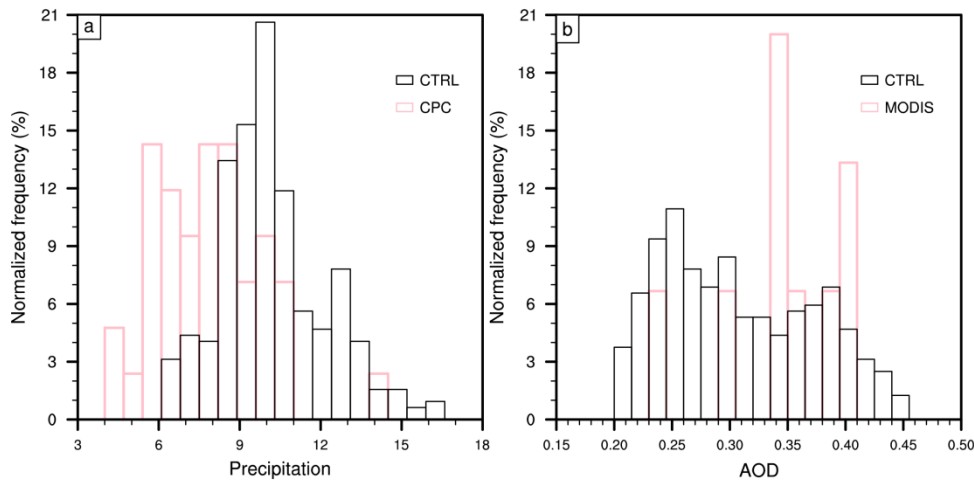

Figure 6. Normalized occurrence frequency of (a) daily precipitation (mm d$^{-1}$) and (b) AOD (unitless) from observations (pink curves, CPC rainfall and MODIS AOD data) and the CTRL experiment (black curves) during the model integration period over the key region of interest (21°–24°N, 111°–116°E) in South China.

Before conducting sensitivity experiments, we evaluate the reliability of the model simulations by comparing the occurrence frequency of precipitation and AOD from the CTRL experiment with observations over the key region of South China (21°–24°N, 111°–116°E), where the primary 8–30-day heavy precipitation event occurred in July 2015 (Fig. 6). The precipitation distribution from the CTRL simulation (black bars in Fig. 6a) generally resembles the observed distribution (pink bars in Fig. 6a). However, CTRL yields a mean precipitation of ~10.2 mm d$^{-1}$ compared with ~7.7 mm d$^{-1}$ in CPC over the key region, corresponding to an overestimation of about 32%. This bias is common in regional climate models, including WRF, and is attributable to limitations in convective cloud and microphysical

parameterizations (Caldwell et al., 2009; Argüeso et al., 2012). Additionally, the CTRL simulation

captures the overall pattern of the observed AOD distribution (Fig. 6b), though it underestimates the mean

AOD by ~35%. This discrepancy is likely due to uncertainties in emission inventories and biases in

meteorological fields (Huang and Ding, 2021). Despite these biases, the CTRL experiment demonstrates

reasonable skill in reproducing the observed precipitation and AOD distributions over the study area.

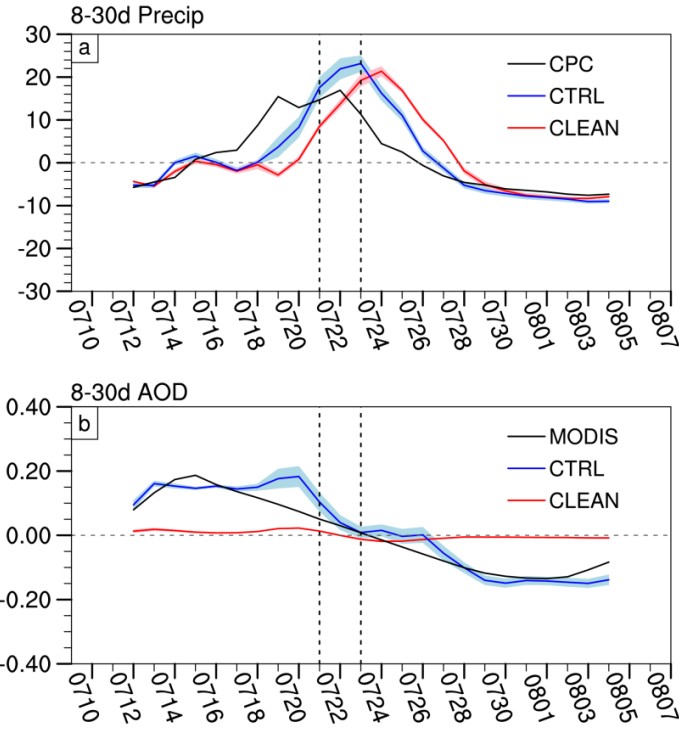

Figure 7. Evolution of (a) precipitation (mm d$^{-1}$) and (b) AOD (unitless) anomalies averaged over the Pearl River Delta

(21°–24°N, 111°–116°E) in the observations (black curves, CPC rainfall and MODIS AOD data), CTRL (blue curves)

and CLEAN (red curves) experiments. Colored shading represents the range of 0.5σ across the six ensemble members.

By comparing rainfall and AOD anomalies—both subjected to a 5-day running mean to remove

high-frequency synoptic signals and focus on the subseasonal timescale—between the CTRL and CLEAN

experiments, the observed relationship between preceding AOD and heavy rainfall is confirmed (Fig. 7).

Notably, the CTRL experiment captures the temporal evolution of precipitation and AOD anomalies

reasonably well, despite slight biases in the timing of their peaks. Quantitatively, the simulated

precipitation anomalies are overestimated by about 6.5 mm d$^{-1}$ relative to CPC observations during the

heavy rainfall case on 21–23 July 2015 (blue curve and shading in Fig. 7a), and the model fails to

reproduce the observed precipitation peak on 19 July. Since the analyses are based on composites aligned

to the precipitation peak (Day 0), the results are not sensitive to small timing offsets of 1–2 days between

model and observations. The robust feature is that positive aerosol anomalies consistently precede

enhanced quasi-biweekly precipitation (black and blue curves in Figs. 7a–b), supporting our main

conclusions and the reliability of the model simulations employed in this study. When AOD is suppressed

in the CLEAN experiment, rainfall amplitude decreases accordingly (red curves in Figs. 7a–b). The most

significant reduction in rainfall amplitude occurs when AOD declines notably between 21–23 July, which will be the focus of the following analyses.

To verify whether aerosols drive rainfall intensification through the aerosol–cloud interactions identified in Sect. 3, we compare these cloud-related variables between the CTRL and CLEAN simulations (Fig. 8). When aerosols are enhanced (transition from CLEAN to CTRL), the key region exhibits an increase in column cloud fraction (Fig. 8a), with distinct positive anomaly centers in the lower troposphere (925 hPa) and upper troposphere (200 hPa). The increase in low-level cloud fraction is attributed to aerosol-induced enhancement in cloud droplet number concentration (figure not shown) and liquid cloud water content (Fig. 8b). Between 700–400 hPa, higher supersaturation promotes increased cloud water formation (Fig. 8b), consistent with observational findings (Fig. 5b). The enhanced upper-level cloud fraction is associated with an increase in ice-phase hydrometeors (Figs. 8d–f), where cloud ice likely forms through cloud water freezing and vapor deposition. Meanwhile, snow growth appears to be driven by depositional processes and the riming of abundant cloud droplets. Although graupel mixing ratios also increase, their magnitude is much smaller compared to the dominant snow mixing ratios. Overall, cold-phase and mixed-phase processes amplify atmospheric rain mixing ratios (Fig. 8c), confirming the observed aerosol-mediated precipitation intensification through enhanced cloud water and cloud ice pathways (Figs. 4–5).

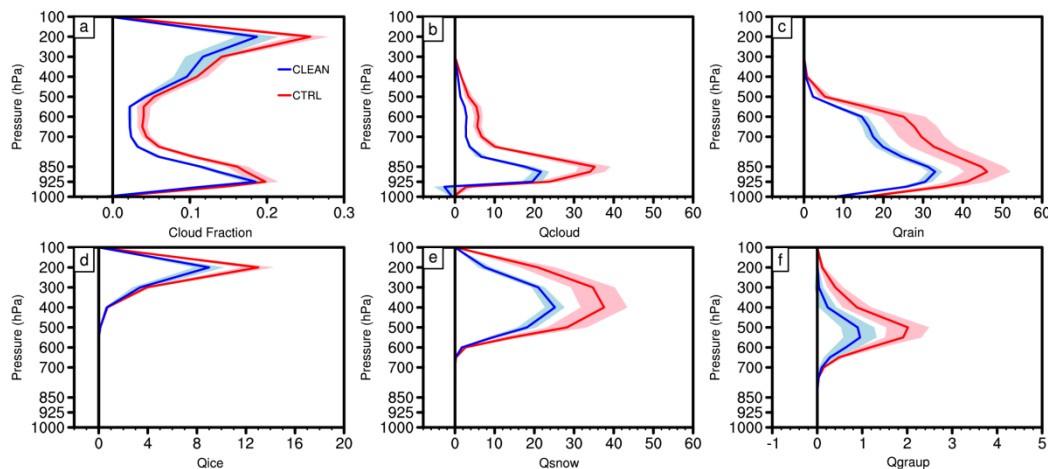

Figure 8. Vertical profiles of (a) cloud fraction anomalies (unitless) and (b)–(f) hydrometeor mixing ratio anomalies ($10^{-6}$ kg kg$^{-1}$) averaged over the key region for the CTRL and CLEAN experiments during 21–23 July 2015: (b) cloud water anomalies, (c) raindrop anomalies, (d) cloud ice anomalies, (e) snow anomalies, and (f) graupel anomalies. Colored shading represents $\pm0.5\sigma$ across the six ensemble members.

To examine the stepwise modulation of heavy rainfall occurrence by aerosol-induced low-level moistening and deep convection development, we compare the vertical profiles of latent heating anomalies, vertical velocity anomalies, and moisture convergence anomalies between the CTRL and CLEAN simulations (Figs. 9a–c). When aerosols are enhanced, the increased latent heating rate exhibits

two distinct centers at 850 hPa and 400 hPa (Fig. 9a), consistent with observed profiles (Fig. 5a). These heating centers correspond to enhanced low-level cloud water and increased high-level cloud ice and snow concentrations (Fig. 8), where latent heat is released through condensation at lower levels and freezing and riming processes at upper levels. Thus, aerosols intensify latent heating through cold-phase and mixed-phase processes, strengthening upward motion throughout the atmosphere (Fig. 9b), similar to the observed patterns (Fig. 5c). Simultaneously, aerosol-induced enhanced low-level moisture convergence anomalies (Fig. 9c) further increase supersaturation and latent heating, amplifying the intensity of rainfall anomalies.

Based on observational diagnostics, we suspect that aerosol–cloud microphysical processes contribute more significantly to rainfall intensification than aerosol–cloud radiative effects (Fig. 5i). To verify this, we analyze the vertical profiles of longwave radiative heating anomalies (Fig. 9d) in the CTRL and CLEAN experiments, along with changes in longwave and shortwave radiation fluxes (Figs. 9e–f). Compared with CLEAN experiment, longwave radiative heating exhibits negligible changes below 925 hPa, but increases by 0.1–0.3 K d$^{-1}$ between 850–500 hPa and above 400 hPa in the CTRL experiment (Fig. 9d), consistent with enhanced cloud fractions (Fig. 8a). To quantify cloud contributions to atmospheric longwave heating, we calculate the longwave cloud-radiative effect using all-sky and clear-sky radiative fluxes based on Eq. (2) (gray bars in Fig. 9e). The development of deep convection triggered by enhanced aerosol emissions reduces OLR by ~10 W m$^{-2}$ at the TOA, increases atmospheric longwave radiation by ~9 W m$^{-2}$, and rises downward longwave radiation by ~1 W m$^{-2}$ at the surface. For shortwave radiation, the increased cloud cover reflects more shortwave radiation back to the TOA (gray bars in Fig. 9f). Additionally, the dominance of scattering aerosols during this rainfall event, as indicated by the high single-scatter albedo, contributes to a small shortwave radiation flux of ~2 W m$^{-2}$ through atmosphere (blue bars in Fig. 9f).

The reduction in latent heating is much larger than that in radiative heating, with values of ~0.8 K d$^{-1}$ for vertical-mean (1000–100 hPa) latent heating compared to ~0.2 K d$^{-1}$ for column longwave heating in the CLEAN experiment relative to CTRL. These modeling results align with observations, showing that aerosol-induced latent heating rates are stronger than aerosol-induced radiative heating rates (Figs. 9a, d). This suggests that aerosol-driven cloud microphysical processes dominate the total heating contribution, while radiative processes play a secondary role (Fig. 4). To test the sensitivity of our results to nesting and spatial resolution, we conducted additional high-resolution experiments with nested domains of 20 km and 4 km (Fig. S6a). These nested simulations consistently demonstrate that aerosols enhance rainfall intensity (Figs. S6b–c) and that aerosol–cloud microphysical effects remain dominant (Fig. S7). Importantly, the ratio of aerosol-induced latent heating to longwave radiative heating remains close to 4:1 (Figs. S7e–f). However, larger biases in AOD and precipitation were evident in the nested

simulation (Figs. S6b–c), likely due to error transmission from the parent grid and accumulated uncertainties introduced at finer resolution during long-term integrations (Baklanov et al., 2014; Wang et al., 2016).

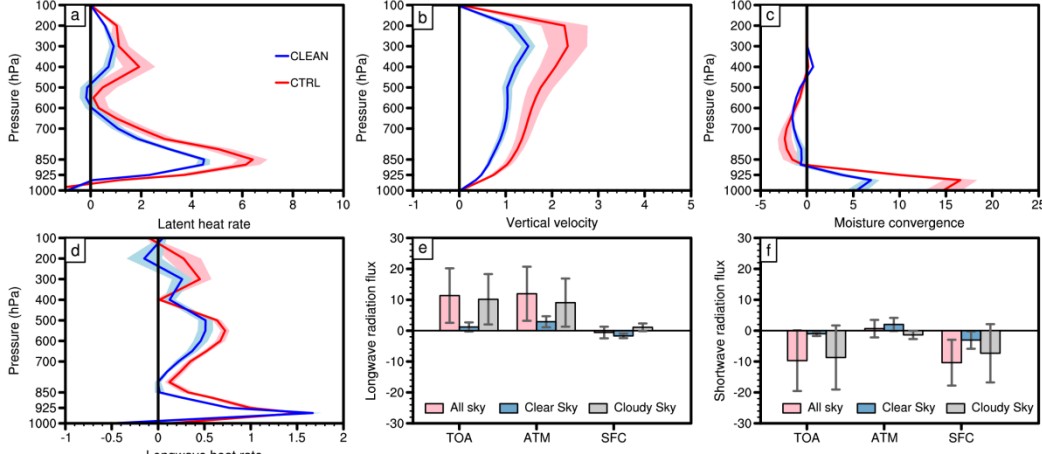

Figure 9. As in Fig. 8, but for the vertical profiles of (a) latent heat flux anomalies (K d$^{-1}$), (b) vertical velocity anomalies (m s$^{-1}$), (c) moisture convergence anomalies ($10^{-2}$ kg kg$^{-1}$ s$^{-1}$), and (d) longwave radiative heating rate anomalies (K d$^{-1}$) in the CTRL and CLEAN experiments. (e) Anomalous longwave radiation fluxes (W m$^{-2}$) averaged over the key region at the top of the atmosphere (TOA), within the whole atmosphere (ATM), and at the surface (SFC) under all-sky (pink bars), clear-sky (blue bars), and cloudy-sky (gray bars) conditions. (f) Similar to (e), but for anomalous shortwave radiative fluxes. Colored bars represent the ensemble mean from six simulations, while whiskers indicate the range of ±0.5σ across the six ensemble members.

## 5 Summary and discussion

South China, a densely populated region, frequently experiences persistent heavy rainfall events that are closely linked to 8–30-day rainfall anomalies. This study investigates how anthropogenic aerosols intensify these quasi-biweekly rainfall anomalies using observational diagnostics and model simulations. Statistical analysis reveals that aerosol concentrations tend to increase approximately four days before the occurrence of rainfall anomalies. The phase-leading increase in AOD is positively correlated with subsequent rainfall intensity, primarily by enhancing moisture convergence, which strengthens deep convection and precipitation. The evolution of moisture sources, cloud properties, and radiative processes suggests that aerosol–cloud microphysical effects play a dominant role in rainfall enhancement, while radiative feedbacks have a secondary influence. These processes are summarized in Fig. 10.

For aerosol–cloud microphysical effects (red and pink sectors in Fig. 10), increased aerosol concentrations enhance CCN activation, promoting the accumulation of low-level cloud water. Latent heat release from the condensation of low-level cloud water induces localized warming, which reduces surface pressure anomalies, strengthens upward motion, and enhances moisture convergence. This

facilitates the uplift of cloud droplets above the freezing level, where additional latent heat release from cold-phase and mixed-phase processes (freezing and deposition) further amplifies ascending motion, leading to stronger deep convection and intensified rainfall anomalies. In addition to microphysical effects, aerosols in southern China can absorb shortwave radiation, while aerosol-induced changes in water vapor and cloud cover also contribute to increased longwave radiation. These radiative processes are illustrated on the right side of Fig. 10 (blue and light blue sectors). The resulting radiative heating provides additional energy for deep convection and enhances moisture flux, further supporting rainfall development.

WRF-Chem experiments for a heavy rainfall event in July 2015, conducted under different aerosol emission conditions, confirm the observational findings. Under CLEAN conditions, where anthropogenic aerosol concentrations are reduced by 90%, precipitation anomalies decease by ~7 mm d$^{-1}$ due to suppressed cold-phase and mixed-phase processes during the heavy rainfall case on 21–23 July 2015 over the key region. In contrast, in the CTRL experiment, increased aerosols appear before enhanced rainfall occurrence, consistent with observations. Enhanced low-level cloud water and mid-to-upper-level ice hydrometeors lead to the formation of dual latent heating centers at 850 hPa and 400 hPa, which drive stronger low-level moisture convergence and column-wide upward motion, ultimately intensifying precipitation anomalies. While longwave cloud-radiative effects marginally increase atmospheric moist static energy, their contribution to precipitation anomalies is relatively weaker. Quantitatively, the column latent heating rate is reduced by ~0.8 K d$^{-1}$ when anthropogenic aerosols are removed, approximately four times greater than the reduction in column longwave radiative heating (~0.2 K d$^{-1}$). These modeling results further corroborate the aerosol-driven intensification of quasi-biweekly precipitation observed in South China.

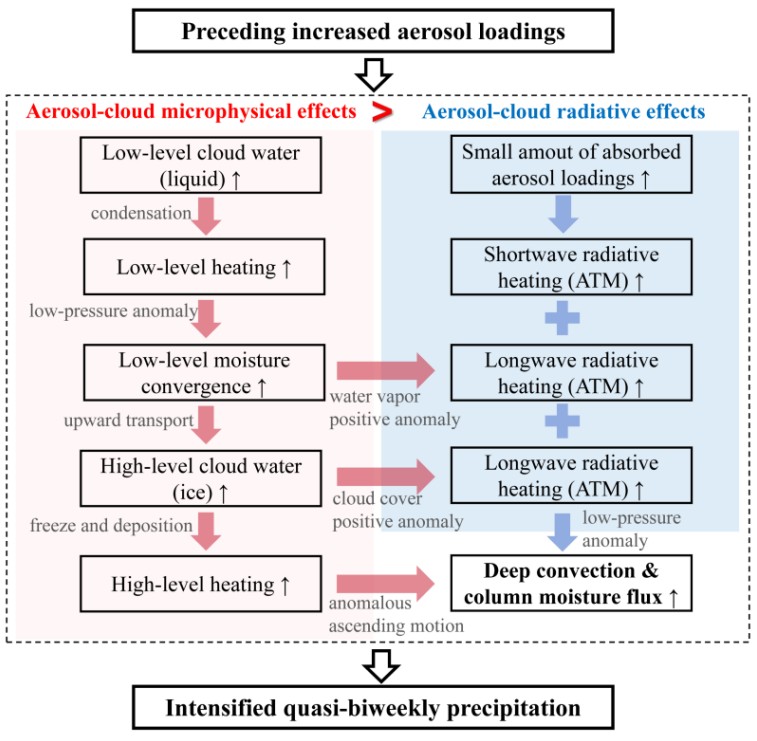

Figure 10. Schematic diagram illustrating the mechanisms of aerosol effects on heavy rainfall events at the 8–30-day timescale. The symbol "↑" denotes the enhancement of the corresponding processes, "ATM" refers to the whole atmosphere, and ">" indicates stronger contributions of the process to heavy precipitation events.

Numerous studies on synoptic and long-term aerosol effects have reported that cloud liquid accumulated by aerosols is converted into ice hydrometeors above the freezing level, thereby invigorating deep convection and intensifying heavy precipitation, the so-called invigoration effect (e.g., Rosenfeld et al., 2008; Li et al., 2011; Hazra et al., 2013; Guo et al., 2014). Aligning with this framework, our observational analysis shows that aerosols similarly enhance cold-phase and mixed-phase cloud development at the 8–30-day timescale, leading to further intensification of heavy precipitation. Our findings contribute to a deeper mechanistic understanding of aerosol–cloud–radiation interactions in modulating precipitation anomalies at the intraseasonal timescale over southern China. Note that the positive contribution of increased AOD to rainfall intensification contrasts with previous studies on the Indian summer monsoon, which reported a link between increased 20–100-day-filtered AOD and the intensification of monsoon breaks (Arya et al., 2021; Surendran et al., 2022). This discrepancy may be attributed to differences in aerosol types and concentrations between the two monsoon regions. In the Indian monsoon region, rainfall suppression has been associated with the indirect effect of dust aerosols through reduced cloud effective radius and decreased precipitation efficiency, whereas nonabsorbing aerosols are more prevalent in southern China (Lee et al., 2007; Huang et al., 2014). Furthermore, Singh et al. (2019) showed that a simultaneous enhancement of dust emissions from West Asia and the Tibetan Plateau intensify the Indian monsoon rainfall with 10–20-day periodicity, but remarkably decrease the spatial scale of the 30–60-day rainfall. Hence, it would be valuable to investigate whether aerosol impacts vary across intraseasonal timescales in southern China, such as the 8–30-day and 20–100-day bands, which we leave for future study.

Given that intraseasonal variability serves as a key source of subseasonal predictability for extreme events (e.g., Wei et al., 2024; Xie et al., 2024), which remains a challenge for both the scientific and operational communities, these results provide valuable insights for improving subseasonal prediction skill by refining aerosol–cloud microphysical processes in prediction models. However, the extent to which aerosols influence subseasonal prediction skill for extreme events requires further investigation. Modulated by intraseasonal convective and circulation anomalies, aerosols exhibit significant variability at the subseasonal timescale (e.g., Tian et al., 2011; Reid et al., 2015; Yu and Ginoux, 2021). However, the factors controlling aerosol loading at the quasi-biweekly timescale in South China remain an open question. Another unresolved issue concerns quantitative attribution. This study provides an initial estimate, but further refined experiments with different aerosol emission scenarios, parameterization schemes, and background conditions are needed for more precise results. In particular, the relative contributions of distinct aerosol pathways to intraseasonal precipitation should be determined by

selectively deactivating aerosol–radiation and aerosol–cloud interactions in WRF-Chem simulations (e.g., Liu et al., 2020; Zhang et al., 2021). These aspects are the focus of our ongoing research.

**Data Availability.** The source codes of the WRF-Chem model are available on the University Corporation for Atmospheric Research (UCAR) website at https://www2.mmm.ucar.edu/wrf/users/download/get_source.html (UCAR, 2025a). The FNL data are available at https: //rda.ucar.edu/datasets/ds083.2/ (NCEP, NWS, NOAA, U.S. DOC, 2000). The biomass burning emission data of FINN version 1.5 can be obtained at https://www.acom.ucar.edu/Data/fire/ (UCAR, 2025b). The MEIC and MIX anthropogenic emissions are available at http://meicmodel.org.cn/ (Tsinghua University, CEADs, CAEP, 2025). The ERA5 data are available at https://cds.climate.copernicus.eu/datasets (last access: 20 September 2025), https://doi.org/10.24381/cds.bd0915c6 (Hersbach et al., 2023a), and https://doi.org/10.24381/cds.adbb2d47 (Hersbach et al., 2023b). MERRA-2 radiation data (https://doi.org/10.5067/Q9QMY5PBNV1T, GMAO, 2015a), aerosol data (https://doi.org/10.5067/KLICLTZ8EM9D, GMAO, 2015b), and meteorological data (https://doi.org/10.5067/QBZ6MG944HW0, GMAO, 2015c) are available at https://disc.gsfc.nasa.gov/datasets (last access: 20 September 2025). The MODIS data is available at https://doi.org/10.5067/MODIS/MOD08_D3.061 (Platnick et al., 2015). The CERES data are from https://doi.org/10.5067/Terra-Aqua-NOAA20/CERES/SYN1degDay_L3.004B (NASA, 2017). The CPC precipitation and OLR data are openly available from NOAA at https://psl.noaa.gov/data/gridded/data.cpc.globalprecip.html (NOAA, 2025a) and https://psl.noaa.gov/data/gridded/data.olrcdr.interp.html (NOAA, 2025b), respectively.

**Author contributions.** PCH and HC conceptualized the research goals and aims. HC performed the analysis and wrote the manuscript draft. AZ and HC ran the simulations. PCH, AZ, and XM reviewed and edited the manuscript.

**Competing interests.** The contact author has declared that none of the authors has any competing interests.

**Acknowledgments.** We appreciate the anonymous reviewers for their constructive comments, which greatly improved the manuscript. This work was supported by the National Natural Science Foundation of China (42225502). We acknowledge the High Performance Computing Center of Nanjing University

of Information Science and Technology for their support of this study. We also thank all the corresponding institutions for providing their data for this study.

**Financial support.** This research has been supported by the National Natural Science Foundation of China (42225502).

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
