# Peer review of "Role of aerosol–cloud–radiation interactions in modulating"

_EGUsphere, 2025_

## Author Comment (AC1)

**Responses to Anonymous Referee #1**

This study investigates the role of aerosol–cloud–radiation interactions in modulating summertime quasi-biweekly rainfall intensity over South China based on both reanalysis data and model simulations, with interesting results provided. Personally, I would like to suggest its acceptance for publication with minor revisions.

**Response:**

We sincerely appreciate your encouraging and constructive comments, which have greatly helped us improve the quality of this study. We also learned a great deal from your suggestions, which provided us with valuable insights into this field. All of your comments have been carefully addressed and the corresponding revisions have been incorporated into the manuscript. Our detailed, point-by-point responses are provided below (in blue).

**Specific comments:**

1. Line 31-33, Recent review studies regarding the aerosol effect on clouds and precipitation could be referred and mentioned, Zhao et al. (2023, doi: 10.1016/j.atmosres.2023.106899) and Li et al. (2019, doi: 10.1029/2019JD030758).

**Response:**

We thank the reviewer for recommending these valuable works, which provide robust insights into aerosol effects on clouds and precipitation. We have carefully studied them and incorporated the relevant findings into the revised Introduction. Please refer to Lines 33–36 in the revised manuscript, as shown below for convenience.

➢ Lines 33–36: "Aerosols influence clouds and precipitation through two primary mechanisms: one involves directly modifying radiation, while the other acts through their role as cloud condensation nuclei (CCN) or ice nuclei (IN) (e.g., Koren et al., 2004; Tao et al., 2012; Li et al., 2016, 2019; Zhu et al., 2022; Zhao et al., 2023; Stier et al., 2024)."

2. Line 35-36, Not always suppressing precipitation, it sometimes enhances precipitation, as indicated by recent studies.

**Response:**

Thank you for this insightful comment. In the revised manuscript, we have modified the description to highlight that aerosol radiative effects can lead not only to suppression but also to enhancement of precipitation, depending on environmental conditions. Please refer to Lines 37–41, as shown below for convenience.

➢ Lines 37–41: "The radiative effect involves the scattering and absorption of solar radiation by aerosols (i.e., the so-called "direct effect"), which commonly leads to atmospheric heating, surface cooling, stabilization of atmospheric stratification, and suppression of precipitation (Bollasina et al., 2011), but can also enhance local or remote precipitation under favorable conditions (e.g., Fan et al., 2015; Zhu et al., 2022; Wei et al., 2023)."

3. Line 37-39, The semi-direct effect often refers the case absorbing aerosols within clouds.

**Response:**

We thank the reviewer for pointing out this important detail regarding the vertical location of absorbing aerosols. In the revised manuscript, we have added this key information to clarify the definition of the semi-direct effect. Please see Lines 41–43, shown below for convenience.

➢ Lines 41–43: "In particular, absorbing aerosols within clouds enhance cloud evaporation, thereby inhibiting cloud and precipitation formation, a phenomenon referred to as the semi-direct effect (Ackerman et al., 2000)."

4. Line 42-43, Actually, there are proposed mechanisms for this invigoration phenomenon, while debates exist.

**Response:**

We thank the reviewer for this important comment. Indeed, the invigoration effect has been widely studied, with several mechanisms proposed — for example, freezing-induced invigoration (Rosenfeld et al., 2008) and condensational invigoration (Fan et al., 2018). At the same time, its occurrence and significance remain debated across different cloud regimes and environmental conditions. To reasonably introduce our research purpose, we have revised the text to explicitly note both the proposed mechanisms and the ongoing debates. Please see Lines 46–48 in the revised manuscript, shown below for convenience.

➢ Lines 46–48: "Additionally, the aerosols can invigorate deep convective cloud through freezing-induced intensification (Rosenfeld et al., 2008) and enhanced condensational heating (Fan et al., 2018), the so-called invigoration effect (Fan et al., 2025), though its significance remains debated qualitatively and quantitatively."

5. Line 66-68, If possible, a short review about the existing studies over South China is appreciated.

**Response:**

Thank you for this helpful suggestion. In South China, previous studies on aerosol–precipitation interactions have largely focused on the start and peak times of diurnal precipitation (Guo et al., 2016; Lee et al., 2016; Sun and Zhao, 2021), mesoscale rainfall intensity variations (Zhang et al., 2020; Xiao et al., 2023a), synoptic-scale rainfall variability (Liu et al., 2020; Guo et al., 2022), and seasonal-to-climatological rainfall changes (Wang et al., 2011; Yang and Li, 2014; Zhu et al., 2023). To better introduce the purpose of our study, we have reorganized the last paragraph of the Introduction and added these discussions. Please see Lines 72–80 in the revised manuscript, as shown below for convenience.

➢ Lines 72–80: "Influenced by active intraseasonal oscillations, persistent heavy precipitation frequently strikes densely populated southeastern China (Hsu et al., 2016), posing increasingly severe threats to socioeconomic development and the

livelihoods of billions. Research on aerosol–precipitation interactions over South China in summer has predominantly examined diurnal precipitation shifts (Guo et al., 2016; Lee et al., 2016; Sun and Zhao, 2021), mesoscale rainfall intensity (Zhang et al., 2020; Xiao et al., 2023a), synoptic-scale rainfall variability (Liu et al., 2020; Guo et al., 2022), and seasonal-to-climatological rainfall changes (Wang et al., 2011; Yang and Li, 2014; Zhu et al., 2023). However, despite the importance of intraseasonal oscillations in regulating regional rainfall, few studies have examined aerosol impacts on intraseasonal variability of rainfall intensity."

6. Line 87, Why do not use the radiation from CERES?

**Response:**

Indeed, the Clouds and the Earth's Radiant Energy System (CERES) provides high-quality Earth radiation budget data. Following this useful suggestion, we collected initial and adjusted radiative fluxes from CERES Synoptic products (abbreviated as CERES-SYN-I and CERES-SYN-A) and compared them with MERRA-2 (Fig. A1). The atmospheric radiative effects associated with intraseasonal rainfall events show highly consistent temporal evolution and magnitudes between the two datasets (Fig. A1a–b). Moreover, the aerosol impacts on the cloud-radiative processes are similar (Fig. A1c). However, CERES-SYN products lack some key parameters: CERES-SYN-I does not provide downward top-of-atmosphere shortwave flux, preventing net shortwave calculations, while CERES-SYN-A omits the pristine-sky condition needed to isolate aerosol effects. For this reason, we used MERRA-2 in the main analysis.

In the revised manuscript, we clarified our rationale for using MERRA-2 (Lines 100–101) and added a supplementary comparison with CERES to demonstrate the robustness of our conclusions (Lines 105–108 and 373). The relevant text is provided below for convenience.

➢ Lines 100–101: "MERRA-2 provides the complete set of variables required for atmospheric radiation and moisture budget quantifications, whereas other reanalyses and observations lack some of these key variables."

- Lines 105–108: "To further reduce uncertainties inherent in reanalyses, we also employed radiative fluxes from Clouds and the Earth's Radiant Energy System Synoptic products (CERES-SYN; Rutan et al., 2015) at 1° resolution, and AOD from the Moderate Resolution Imaging Spectroradiometer (MODIS) Collection 6 Level-3 aerosol product onboard the Terra satellite (Levy et al., 2013) at 1° resolution."

- Line 373: "This behavior is consistent with estimates from CERES-SYN (Fig. S4)."

[Figure]

Figure A1. (a) Composite evolution of 8–30-day longwave and shortwave cloud radiative effects (Cld_LW, magenta curve; Cld_SW, pink curve; W m⁻²) calculated from Eq. (2), derived from CERES-SYN-A (solid curves), CERES-SYN-I data (dash-dotted curves), and MERRA-2 (dashed curves), associated with High AOD–Strong Precipitation (HA–SP) events. Day 0 denotes the peak of rainfall events, while negative and positive values on the *x*-axis indicate days before and after the peak, respectively. (b) and (c) are similar to (a), but represent the composite results for Low AOD–Strong Precipitation (LA–SP) events and the differences between HA–SP and LA–SP events, respectively. In panel (c), the periods when their differences with statistically significant differences at the 90% confidence level are marked by gray asterisks.

7. Line 93-96, Similarly, why do not use CloudSat/Calipso observations?

**Response:**

We appreciate this constructive suggestion. CloudSat provides high-quality cloud water products, but large temporal gaps prevent its use for analyzing continuous sequences of aerosol–cloud–precipitation interactions at intraseasonal timescales. For this reason, we used reanalyses (MERRA-2 and ERA5) to examine vertical cloud water structures. To ensure their reliability, we compared reanalysis cloud water content profiles with CloudSat 2B-CWC-RO products (Austin et al., 2009). Because of known uncertainties in Cloud Profiling Radar (CPR) retrievals within ~0.5–0.7 km above the

surface (Stephens et al., 2008), we excluded CloudSat data below ~0.7 km from the comparison (Zhang et al., 2015). As shown in Fig. A2, ERA5, MERRA-2, and CloudSat capture similar vertical distributions of liquid and ice cloud water content over South China, with ice peaking in the upper troposphere and liquid showing two maxima at middle and lower levels. These features are similar to previous CloudSat-based studies in this region (Yang and Wang, 2012; Zhang et al., 2015), supporting the use of reanalyses in our study.

In the revised manuscript, we clarified this rationale in Lines 113–116, as shown below.

➢ Lines 113–116: "Although CloudSat provides three-dimensional cloud products (Austin et al., 2009), substantial temporal gaps prevent its use for analyzing continuous sequences of aerosol–cloud–precipitation interactions at intraseasonal timescales. Thus, three-dimensional liquid and ice cloud water contents were instead taken from MERRA-2 and ERA5 to evaluate vertical cloud structures."

[Figure]

Figure A2. Vertical profiles of mean (a–c) ice cloud water content and (d–f) liquid cloud water content from (a, d) ERA5, (b, e) MERRA-2, and (c, f) CloudSat over South China during boreal summer (May–September) 2006–2020. Units: g m$^{-3}$

8. Line 134-136, To be fair, limitations for model studies should also be acknowledged.

**Response:**

We thank the reviewer for this important suggestion. Indeed, while WRF-Chem simulation is a valuable tool to disentangle causal links between aerosols, clouds, and precipitation, they inevitably involve uncertainties, particularly related to the choice of physical parameterizations, emission inventories, and initial and boundary conditions. We have briefly acknowledged these limitations in Section 2.3 (Lines 154–157), and further discussed them in Section 4 when presenting the model results. The revised text is provided below.

➢ Lines 154–157: "To address this, we conducted a series of experiments using the WRF-Chem version 4.2.2 (Grell et al., 2005; Fast et al., 2006) to support the observed mechanisms responsible for aerosol impacts on clouds and precipitation, although uncertainties remain due to the dependence on emission inventories, physical parameterizations, and initial and boundary conditions."

9. Line 144-145, Could this nudging reduce/remote some effects from aerosol-meteorology interactions? And what will this affect the analysis results?

**Response:**

Thank you for raising this important point. In our experiments, grid nudging was applied only during the spin-up period to better initialize meteorology, while aerosol–meteorology interactions were analyzed during the subsequent free-running period. This design ensures that nudging does not interfere with the main analysis. We recognize that nudging can partially suppress aerosol impacts on model dynamics (He et al., 2017), so we performed a sensitivity experiment without nudging. The results show consistent aerosol-induced precipitation enhancement (Fig. A3a), confirming that our conclusions are not sensitive to the nudging procedure.

In the revised manuscript, we clarified the nudging setup and discussed its potential impacts on our conclusions (Lines 166–170), as provided below.

➤ Lines 166–170: "To better reproduce the observed circulation and aerosol pattern, grid analysis nudging is applied only during the spin-up period (Abida et al., 2022), allowing meteorological fields to freely interact with aerosols during the analysis period. While nudging could potentially constrain aerosol feedbacks (He et al., 2017), our sensitivity tests confirm that it does not affect the main conclusions (figure not shown)."

[Figure]

Figure A3. (a) Evolution of precipitation (mm d⁻¹) and (b) AOD (unitless) anomalies averaged over the Pearl River Delta (21°–24°N, 111°–116°E) in the observations (black curves, CPC rainfall and MODIS AOD data), CTRL (blue solid curves) and CLEAN (red solid curves) of no-nudging simulations, as well as CTRL experiment of nudging simulation (blue dashed curves). All simulations are initialized on 9 July 2015.

10. Line 168-170, Why do the authors use so long time as spin-up, instead of 12 or 24 hours as used by many studies?

**Response:**

We thank the reviewer for this insightful question. In this study, a longer spin-up period was chosen to ensure that locally emitted aerosols became sufficiently mixed and reached a quasi-equilibrated distribution before the analysis period, consistent with earlier WRF-Chem applications to aerosol–meteorology interactions (e.g., Zhu et al., 2022; Wei et al., 2023; Agarwal et al., 2024). To test sensitivity, we performed ensemble simulations with spin-up times ranging from 1 to 6 days, including a 24-hour spin-up. The ensemble results show consistent aerosol-induced precipitation responses, with small ensemble-mean uncertainties, indicating that our conclusions are not sensitive to the choice of spin-up length.

We also recognize that long integrations may accumulate model biases. To minimize this potential impact, grid nudging was applied during the spin-up period, which helps constrain large-scale circulation and reduce drift.

In the revised manuscript, we clarified the rationale for using a multi-day spin-up (Lines 196–199). Please see below for your convenience.

➤ Lines 196–199: "To allow locally emitted aerosols to become sufficiently mixed and reach a quasi-equilibrated distribution, we adopted spin-up times of 1–6 days, consistent with previous studies (e.g., Zhu et al., 2022). The first few days of each run (4–9 July) are discarded, and the analysis focuses on 10 July–6 August 2015."

11. Line 235-236, One more 50-year observation based climatological study by Su et al. (2020, doi: 10.3390/atmos11030303) is worthy to refer here.

**Response:**

We apologize for overlooking this important reference and thank you for bringing it to our attention. After reading it, we have cited this paper in Lines 262–265. For your convenience, the details are provided below.

➤ Lines 262–265: "This behavior aligns with previous studies at the synoptic and decadal timescales (Wang et al., 2011; Yang et al., 2018; Su et al., 2020; Shao et al., 2022; Xiao et al., 2023b), which emphasize that the aerosols tend to suppress light rainfall while enhancing heavy convective precipitation."

12. Line 316, cloud ice particles.

**Response:**

Thank you for pointing this out. The term has been corrected to "cloud ice particles" in the revised manuscript (Line 358).

13. Line 380, I am not sure if we can use "verification" or not since these are not

observations, but model simulations, while we could say "support".

**Response:**

We agree with the reviewer that "support" is a more precise term in this context, since the results are based on model simulations rather than direct observations. Accordingly, we have replaced "verification" with "support" in the revised manuscript (Line 429).

**References**

[revised manuscript text omitted]

---

## Author Comment (AC2)

**Responses to Anonymous Referee #2**

This manuscript presents a thorough investigation of aerosol effects on 8–30-day rainfall anomalies, integrating observational datasets and WRF-Chem simulations. The study addresses a timely and relevant topic with important implications for subseasonal precipitation prediction.

While the manuscript is generally well-structured, several sections would benefit from clearer explanations of methodological assumptions, diagnostic frameworks, and limitations.

Many analyses remain qualitative. For improved scientific rigour and clarity, the authors are encouraged to report more quantitative metrics throughout (e.g., % changes, W m⁻², correlation coefficients, σ-standardised anomalies). Figs. should be enlarged where necessary and include confidence intervals or error bars where applicable to support statistical interpretation.

**Response:**

We deeply appreciate the reviewer's constructive comments and useful suggestions. We have carefully considered the concerns regarding methodological clarity, quantitative rigor, and expression quality, and have refined the manuscript accordingly. Our detailed point-by-point responses are provided below (in blue).

**Specific comments:**

1. Lines 26–27: Could the authors clarify how dominant the microphysical effect is compared to radiative forcing in quantifiable terms (e.g., W/m², % contribution)?

**Response:**

We thank the reviewer for this important suggestion. In our analysis, aerosol–cloud microphysical effects are represented by aerosol-induced latent heating ($Q_2$, discussed in comment #16), while aerosol–radiative effects are represented by dominant longwave cloud-radiative heating. Observations show that aerosol-induced latent heating reaches 40 W m⁻², about seven times larger than the corresponding

radiative heating (2–5.5 W m⁻²). As noted in our response to comment #17, model simulations yield a comparable ratio of roughly four (~0.8 K d⁻¹ vs. ~0.2 K d⁻¹). These results consistently demonstrate the dominant role of microphysical effects in modulating quasi-biweekly precipitation intensity.

We emphasize, however, that these estimates are based on indirect evidence. More precise quantitative partitioning of aerosol–radiation versus aerosol–cloud contributions would require targeted sensitivity experiments in which each pathway is selectively deactivated (e.g., Liu et al., 2020; Zhang et al., 2021).

In the revised manuscript, we added the quantitative assessment in the Abstract (Lines 28–29) and expanded the discussion of limitations (Lines 610–615), as shown below.

➢ Lines 28–29: "Quantitatively, aerosol-induced latent heating exceeds aerosol-induced longwave radiative heating by a factor of ~4–7 in both observations and model simulations."

➢ Lines 610–615: "Another unresolved issue concerns quantitative attribution. This study provides an initial estimate, but further refined experiments with different aerosol emission scenarios, parameterization schemes, and background conditions are needed for more precise results. In particular, the relative contributions of distinct aerosol pathways to intraseasonal precipitation should be determined by selectively deactivating aerosol–radiation and aerosol–cloud interactions in WRF-Chem simulations (e.g., Liu et al., 2020; Zhang et al., 2021)."

2. Lines 46–48: Could the authors elaborate on what these uncertainties are? How do you isolate aerosol-induced variability (especially AOD) from synoptic-scale meteorological variability that might independently influence precipitation?

**Response:**

Thank you for this thoughtful question. We apologize for the earlier imprecise phrasing. By "uncertainties in the aerosol–precipitation relationship," we refer to the diverse ways aerosols interact with precipitation. This diversity arises from spatial heterogeneity in topography, moisture, and aerosol properties, as well as temporal

variability in synoptic system intensity and movement, aerosol emission accumulation, and multi-scale interactions.

To isolate aerosol-induced variability, we identify high- and low-emission events and attribute their differences to aerosol influences. Although meteorological fields and aerosols often covary, when AOD anomalies are found to lead circulation and precipitation fields, aerosols are considered potentially active in modulating circulation and rainfall. Because AOD could be influenced by humidity, we further validated our findings with the independent gridded $PM_{2.5}$ dataset (ChinaHighPM2.5; Wei et al., 2020, 2021). The composites show that enhanced $PM_{2.5}$ concentrations precede 8–30-day heavy precipitation events by ~4 days (Fig. B1), confirming that the observed leading phase of aerosols is robust regardless of whether AOD or $PM_{2.5}$ is used.

In the revised manuscript, we clarified these in both the Introduction (Lines 51–53) and Results (Lines 237–240) sections, as shown below for your convenience.

➢ Lines 51–53: "Diversity in the aerosol–precipitation relationship arises across regions and timescales due to variations in dynamic and thermodynamic conditions, aerosol properties, and multi-scale interactions."

➢ Lines 237–240: "Validation with long-term reconstructed gridded datasets of particulate matter smaller than 2.5 μm ($PM_{2.5}$) confirms these findings (Fig. S1c), indicating that the observed leading phase of aerosols is robust, regardless of whether AOD or $PM_{2.5}$ is used as the proxy for aerosol concentrations."

[Figure]

Figure B1. Evolution of 8–30-day precipitation (red curve, mm d$^{-1}$) and 8–30-day $PM_{2.5}$ (blue curve, μg m$^{-3}$) associated with quasi-biweekly precipitation events, defined as periods with positive 8–30-day rainfall anomalies over South China. Day 0 denotes the peak of rainfall events, while negative and positive values on the *x*-axis indicate days before and after the peak, respectively.

3. Lines 82–86: The study uses MERRA-2 for aerosol variables. Considering the availability of higher-resolution datasets such as ERA5 (0.25°) and CAMS, could the authors justify this choice? Would ERA5+CAMS provide better representation of subseasonal aerosol‑meteorological interactions over South China?

**Response:**

We appreciate this thoughtful suggestion. We addressed your two concerns as follows.

(1) While CAMS and ERA5 indeed provide higher spatial resolution for AOD (0.75°) and meteorological variables (0.25°), we primarily used MERRA-2 because it offers a complete and internally consistent set of variables needed for both radiative and moisture budget quantifications. Some of these key variables are not available in ERA5, which limits their direct use for our full diagnostic framework.

(2) Nevertheless, we agree that higher-resolution datasets may better capture regional variability. Following your suggestion, we conducted additional analyses using CAMS AOD and ERA5 (0.25°). As shown in Fig. B2, enhanced AOD consistently precedes 8–30-day heavy precipitation events by about two days, and the antecedent AOD anomalies remain significantly correlated with subsequent precipitation events exceeding the $0.25\sigma$ threshold. Furthermore, Fig. B3 demonstrates that the aerosol impacts on quasi-biweekly precipitation, moisture and cloud-radiative processes derived from CAMS AOD and high-resolution ERA5 are highly consistent with those obtained from MERRA-2. These results confirm that our main conclusions are robust across multiple datasets.

In the revised manuscript, we clarified the rationale for selecting MERRA-2 while also incorporating high-resolution ERA5 and CAMS analyses to evaluate robustness (Lines 101–105). The ERA5- and CAMS-based results are included in the Supplementary Materials (Figs. S1–S3 and Fig. S5), with relevant discussions added in Lines 236–237, 296–298, 330–332, 404–405, and 420–423.

➢ Lines 101–105: "Even so, to ensure that our conclusions are not dataset-dependent, we compared budget analysis results with those derived from the fifth-generation

European Center for Medium-Range Weather Forecasts (ECMWF) atmospheric reanalysis (ERA5; Hersbach et al., 2020), which has a spatial resolution of 0.25° × 0.25° and 37 vertical levels."

➢ Lines 236–237: "The consistent evolution of AOD from MODIS and ERA5 datasets (Figs. S1a–b) demonstrates the robustness of intraseasonal AOD variations associated with heavy rainfall events."

➢ Lines 296–298: "Composite patterns of convection and circulation during the early and peak phases of HA–SP and LA–SP events, along with their differences, are shown in Fig. 3 based on MERRA-2 data, and are consistent with those derived from ERA5 (Fig. S2)."

➢ Lines 330–332: "Overall, the key moisture processes associated with quasi-biweekly precipitation events and their modulation by aerosols are consistently shown in the ERA5 reanalysis (Figs. S3a–c)."

➢ Lines 404–405: "The relative contributions and quantitative ratios between the two variables are similarly shown in the ERA5 data (Figs. S3d–f)."

➢ Lines 420–423: "The enhanced conversion of supercooled cloud water to ice hydrometeors through freezing and deposition further releases latent heat, reinforcing vertical motion and convection. These processes ultimately intensify rainfall anomalies, a finding that is also confirmed by ERA5 (Fig. S5)."

[Figure]

Figure B2. (a) Evolution of 8–30-day precipitation (red curve, mm d⁻¹) and 8–30-day CAMS AOD (blue curve, unitless) associated with quasi-biweekly precipitation events, defined as periods with positive 8–30-day rainfall anomalies over South China. Day 0 denotes the peak of rainfall events, while negative and positive values on the *x*-axis indicate days before and after the peak, respectively. (b) Scatterplot of rainfall intensity (i.e., 8–30-day-filtered rainfall anomalies at Day 0; *x*-axis, mm d⁻¹) versus preceding AOD intensity (represented by the peak value of 8–30-day-filtered AOD anomalies during Days –6 to –1; *y*-axis, unitless), with both variables normalized by their

climatology. The red line represents the linear regression fitted to all cases, while the blue line corresponds to events where rainfall intensity exceeds 0.25σ. The correlation coefficients between the two variables are shown in the bottom right corner, with two asterisks indicating significance at the 95% confidence levels.

[Figure]

Figure B3. (a) As in Fig. B2a, but for the composite evolution of 8–30-day (a) precipitation (red curve; left $y$-axis, mm d$^{-1}$), CAMS AOD (blue curve; right $y$-axis in blue, unitless), and individual moisture budget terms based on 0.25°-resolution ERA5 data (various colored curves representing each budget term in Eq. (1); left $y$-axis, mm d$^{-1}$) associated with High AOD–Strong Precipitation (HA–SP) events. (d) As in (a), except that (d) shows 8–30-day column-integrated latent heat heating ($Q_2$; green curve; left y-axis in green, W m$^{-2}$), 8–30-day longwave and shortwave cloud radiative effects (Cld_LW, magenta curve; Cld_SW, pink curve) calculated from Eq. (2) (right y-axis, W m$^{-2}$). (b, e) and (c, f) are similar to (a, d), but represent the composite results for Low AOD–Strong Precipitation (LA–SP) and the differences between HA–SP and LA–SP events, respectively. In panels (c) and (f), only terms with statistically significant differences at the 90% confidence level are shown, with significant periods marked by gray asterisks.

4. Lines 94–99: ERA5 is used for cloud and circulation diagnostics. Could the authors explain why a single data source (e.g., ERA5+CAMS) was not adopted consistently throughout the study to avoid discrepancies in spatial/temporal resolution or reanalysis biases? Could the authors clarify why ERA-Interim was chosen as input for WRF-Chem, rather than ERA5, which offers significantly improved spatial, temporal, and vertical resolution?

**Response:**

Thank you for raising these important comments. We address the two issues below.

(1) Choice of datasets: As noted in our earlier reply, neither ERA5 nor CAMS provides the full set of radiative fluxes required for budget analysis, particularly radiative fluxes under clear-sky, no-aerosol conditions. This limitation is the main reason why MERRA-2 was used for radiative and moisture budget quantifications in the main text. To ensure that our results are not dataset-dependent, we additionally analyzed ERA5 and CAMS data, which are now presented in Figs. S1–S3 and Fig. S5, with accompanying discussions (see also our response to comment #3).

(2) WRF-Chem input data: To clarify, our simulations used NCEP-FNL rather than ERA-Interim as initial and boundary conditions for WRF-Chem. We also conducted parallel experiments using ERA5 data at 0.25° resolution and 3-hourly intervals. These tests show that the CTRL experiment with ERA5 input introduces larger biases in AOD magnitude and temporal evolution compared with both observations and the NCEP-FNL–driven simulation (Fig. B4b). Moreover, although ERA5 input improves the timing of precipitation evolution, it overestimates precipitation anomalies during 26–28 July (Fig. B4a). Similar biases have been reported in previous WRF studies using ERA5 versus NCEP-FNL as input (e.g., Parra, 2022; Chanchal and Singh, 2024; Fu et al., 2025). Based on these results, we retained NCEP-FNL as the WRF-Chem input dataset. This rationale is now clarified in the revised manuscript (Lines 165–166).

➢ Lines 165–166: "Simulations using NCEP-FNL data as WRF-Chem input better capture the AOD evolution compared to those driven by higher-resolution ERA5 data (figure not shown)."

[Figure]

Figure B4. Evolution of (a) precipitation (mm d$^{-1}$) and (b) AOD (unitless) anomalies averaged over the Pearl River Delta (21°–24°N, 111°–116°E). Observations are shown by black curves (CPC precipitation and MODIS AOD), while CTRL experiments are driven by ERA5 data (blue solid curves) and NCEP-FNL data (blue dashed curves). All simulations are initialized on 9 July 2015.

5. Line 139: Is the 20 km horizontal resolution sufficient to resolve mesoscale convection associated with aerosol–precipitation feedbacks? Were nested domains with higher resolution tested?

**Response:**

Yes, we agree that cloud-resolving scales (≤4 km) are generally required to explicitly resolve mesoscale convection and aerosol–cloud microphysical processes. Following Liu et al. (2020), we therefore conducted additional high-resolution experiments with nested domains of 20 km and 4 km for July–August 2015. These simulations confirm that aerosols enhance rainfall intensity and that aerosol–cloud microphysical effects remain dominant relative to aerosol-induced longwave radiative effects (Figs. B5–B6). Quantitatively, the vertical-mean (1000–100 hPa) latent heating difference between the CTRL and CLEAN experiments is ~0.343 K d$^{-1}$, about four times greater than the corresponding longwave radiative heating (~0.087 K d$^{-1}$). This relative contribution is consistent with the single-domain simulation (see our response to comment #17). However, the nested configuration did not significantly improve performance relative to the single-domain 20 km simulation: it delayed the precipitation peak by three days and underestimated AOD magnitude, likely due to error transmission from the parent grid and accumulated uncertainties introduced at finer resolution during long-term integrations (Baklanov et al., 2014; Wang et al., 2016).

By contrast, the 20 km simulations capture the temporal evolution of AOD and rainfall over South China reasonably well, providing sufficient fidelity to support our observational analysis without introducing additional biases. Therefore, in the revised manuscript we present the 20 km results in the main text and include discussion of the high-resolution nested experiments in Lines 526–534 (with supporting figures in the Supplementary Material), as shown below for convenience.

➢ Lines 526–534: "To test the sensitivity of our results to nesting and spatial resolution, we conducted additional high-resolution experiments with nested domains of 20 km and 4 km (Fig. S6a). These nested simulations consistently demonstrate that aerosols enhance rainfall intensity (Figs. S6b–c) and that aerosol–

cloud microphysical effects remain dominant (Fig. S7). Importantly, the ratio of aerosol-induced latent heating to longwave radiative heating remains close to 4:1 (Figs. S7e–f). However, larger biases in AOD and precipitation were evident in the nested simulation (Figs. S6b–c), likely due to error transmission from the parent grid and accumulated uncertainties introduced at finer resolution during long-term integrations (Baklanov et al., 2014; Wang et al., 2016)."

[Figure]

Figure B5. (a) WRF-Chem nested model domains with horizontal resolutions of 20 km (domain 1) and 4 km (domain 2). (b) Evolution of precipitation anomalies (mm d⁻¹) and (c) AOD anomalies (unitless) averaged over the Pearl River Delta (21°–24°N, 111°–116°E). Black curves denote observations (CPC rainfall and MODIS AOD), while blue and red solid curves represent CTRL and CLEAN experiments of the nested-domain simulation, respectively. The CTRL experiment of the single-domain simulation is also shown (blue dashed curves). All experiments are initialized on 9 July 2015

[Figure]

Figure B6. Vertical profiles of (a)–(d) hydrometeor mixing ratio anomalies (10⁻⁶ kg kg⁻¹) averaged over the key region for the CTRL and CLEAN experiments during 23–26 July 2015 in the nested simulations: (a) raindrop anomalies, (b) cloud ice anomalies, (c) snow anomalies, and (d) graupel anomalies. (e)–(f) As in (a), but for the vertical profiles of (e) latent heat rate anomalies (K d⁻¹), and (f) longwave heat rate anomalies (K d⁻¹).

6. Lines 159–160: WRF-Chem is known to underestimate dust emissions, particularly in East and Southeast Asia. Were adjustments made (e.g., emission scaling or tuning) to account for this bias?

**Response:**

Thank you for this insightful comment. As you indicated, WRF-Chem is known to underestimate dust aerosol concentrations in Asia (Zhao et al., 2020; Zhu et al., 2022). In our case, however, the simulated heavy rainfall event occurred during 19–24 July 2015, when dust loading was minimal in our model domain (Fig. B7). Over South China, dust AOD was ~0.01 (Fig. B7a), AI values were <0 (Fig. B7b), and SSA exceeded 0.95 (Fig. B7c). These indicators confirm that aerosols were dominated by scattering types (Lee et al., 2010; Wilcox, 2012), with dust exerting little influence on our results.

Since dust effects were negligible in our case, no emission scaling or tuning was applied. We acknowledge, however, that future work should incorporate such adjustments to improve dust representation in cases where dust cannot be neglected. We have clarified this point in the revised manuscript (Lines 185–188).

➢ Lines 185–188: "Although the GOCART scheme may underestimate dust aerosol concentrations in Asia (Zhao et al., 2020), no emission scaling or tuning was applied in this study because our simulations focus on anthropogenic aerosols and summer rainfall in South China, where dust emissions contribute minimally."

[Figure]

Figure B7. Spatial distributions of (a) dust AOD based on MERRA-2 (unitless), (b) Aerosol Index (AI; unitless) derived from the Ozone Monitoring Instrument (OMI) during 19–24 July 2015, and (c) aerosol single-scattering albedo (SSA; unitless) derived from merged OMI satellite products and AERONET ground-based observations (Dong et al., 2025) during July 2015.

7. Lines 203–205: The text implies that precipitation anomalies precede the composite reference point, but Fig. 2a suggests that AOD anomalies occur ~6 days prior to rainfall anomalies. Could the timing be clarified?

**Response:**

The reviewer is correct that positive AOD anomalies occur about six days before the rainfall peak, which is defined as Day 0, while rainfall anomalies begin to increase about three days before the peak. To avoid confusion, we clarified the definition of Day 0 and the timing of anomalies in both the text and the figure caption."

➢ Lines 232–233: "Composite analysis shows that AOD exhibits positive anomalies beginning about six days before the peak of 8–30-day rainfall events (Day 0; red curve in Fig. 2a; blue curve for AOD)."

➢ Lines 281–282: "Day 0 denotes the peak of rainfall events, while negative and positive values on the *x*-axis indicate days before and after the peak, respectively."

8. Line 223: The correlation between AOD and rainfall is relatively weak (r = 0.25). Could the observed relationship be influenced by shared drivers such as regional circulation patterns?

**Response:**

We agree with the reviewer that the correlation between AOD and rainfall (r = 0.25) is modest, reflecting the complexity of rainfall generation processes. Statistically, the correlation is significant given the large sample size (n = 87), indicating that higher AOD tends to precede stronger rainfall anomalies. Physically, this modest value suggests that rainfall intensification is controlled by multiple drivers, including circulation anomalies that may co-vary with or act independently of AOD variations. To address this, we further examined how different AOD conditions and associated circulation anomalies influence rainfall intensification (Fig. 3).

In the revised manuscript, we clarified this point in Lines 266–271, as reproduced below.

➢ Lines 266–271: "The statistical results in Fig. 2 indicate that AOD anomalies have a more pronounced impact on 8–30-day precipitation events exceeding a certain

intensity threshold (e.g., 0.25σ), with stronger antecedent AOD anomalies leading to amplified subsequent precipitation anomalies, although the correlation coefficients are modest (r=0.25–0.39). These modest values highlight the complexity of rainfall intensification mechanisms, which involve circulation anomalies that may be induced by, or independent of, AOD variations."

9. Lines 256–258: The term "stronger antecedent AOD anomalies" would benefit from quantification. What time window defines "antecedent"? Could the authors specify how much precipitation amplification is observed (e.g., % increase, mm/day)?

**Response:**

We thank the reviewer for this helpful comment and agree that both clarification and quantification are needed.

(1) Time window: In the caption of Fig. 2 (Line 284), we have specified that "antecedent AOD anomalies" refer to the 8–30-day-filtered AOD anomalies during Days –6 to –1. To emphasize this, we also reiterate the definition in the description of Fig. 2b in the main text (Lines 243–244).

(2) Quantification: The suggestion to quantify rainfall amplification associated with increased AOD is excellent. We have incorporated this into our discussion of the evolution of AOD and rainfall anomalies in Fig. 4 (Lines 324–328), as shown below.

➢ Lines 243–244: "Fig. 2b examines how subsequent rainfall anomalies vary with the amplitude of preceding AOD anomalies during Days –6 to –1, considering all 8–30-day rainfall events."

➢ Lines 324–328: "An increment of ~0.1 in AOD at Day –4 slightly precedes the enhancement of moisture convergence (blue and green curves in Fig. 4c), indicating that the aerosols could play roles in moistening process, which may subsequently lead to intensification of 0.7–1 mm d$^{-1}$ in rainfall during Days –2 to 4 (red curve in Fig. 4c). At Day 0, the intensity of quasi-biweekly precipitation is increased by ~20%."

10. Line 264: Why were LA–SP cases with −0.4σ < AOD < 0 excluded? Could this exclusion skew the composite comparison?

**Response:**

Thank you for this important question. In order to differentiate clean and polluted conditions in observations, we categorized samples into two event types using percentile-based thresholds (e.g., Zhou et al., 2020; Sun and Zhao, 2021; Zhu et al., 2024). As shown in Fig. 2b, cases with AOD anomalies close to the climatological mean (i.e., near zero) exhibit only minimal differences between high- and low-AOD composites. If such cases were included, the contrast between the two categories would be too weak to meaningfully represent aerosol effects. To ensure that the two categories reflect clearly distinct high- and low-loading conditions while maintaining balanced and sufficient sample sizes, we excluded samples with $-0.4\sigma < \text{AOD} < 0$. This effectively retains the top and bottom 40% of AOD anomalies while discarding the middle 20%, which carry the least signal.

In the revised manuscript, we clarified this rationale in Lines 275–277, as provided below.

➢ Lines 275–277: "To distinguish clean and polluted conditions and ensure balanced sample sizes, LA–SP cases with AOD anomalies between $-0.4\sigma$ and 0 (approximately 40th–60th percentiles) were excluded (Fig. 2b)."

11. Lines 289–298: Please provide numerical values (e.g., means, ranges) to support the interpretation of rainfall or cloud anomalies in this section.

**Response:**

We appreciate this valuable suggestion. To strengthen our interpretation, we revised this section to include quantitative statistics. The updated text is provided in the revised manuscript (Lines 316–328) and copied below for convenience.

➢ Lines 316–328: "In both cases, moisture convergence ($-\langle q\nabla \cdot V\rangle'$, green curves in Figs. 4a–b) demonstrated a growth of 2–3 mm d⁻¹ at Day 0, accounting for ~50% of the positive rainfall anomalies (red curves in Figs. 4a–b) and serving as the primary moisture source. Moreover, moisture convergence also explains the differences in rainfall amplitude between HA–SP and LA–SP events (green curve in Fig. 4c), which exhibited a significant increase of 1–1.8 mm d⁻¹ during Days −4 to 0. The moisture sink associated with latent heating is in phase with rainfall and

offsets the moisture source from convergence with a reduction of 0.9–1.6 mm d⁻¹ during Days −3 to 1 (cyan curve in Fig. 4c). Nevertheless, their combined effect still contributes positively to heavy rainfall occurrence (red curve in Fig. 4c). The temporal evolution of these key terms reveals their sequential influence. An increment of ~0.1 in AOD at Day −4 slightly precedes the enhancement of moisture convergence (blue and green curves in Fig. 4c), indicating that the aerosols could play roles in moistening process, which may subsequently lead to intensification of 0.7–1 mm d⁻¹ in rainfall during Days −2 to 4 (red curve in Fig. 4c). At Day 0, the intensity of quasi-biweekly precipitation is increased by ~20%."

12. Lines 298–300: Could the authors provide an example or reference to clarify what is meant by "other processes"?

**Response:**

Thank you for pointing out this unclear description. In the original manuscript, "other processes" referred to horizontal moisture advection and vertical moisture flux. We have now made it explicit in the revised manuscript (Lines 328–330), as shown below for convenience.

➢ Lines 328–330: "The horizontal moisture advection and vertical moisture flux (magenta curves and pink curves in Figs. 4a–b) make relatively minor contributions, and their differences between HA–SP and LA–SP events are not statistically significant."

13. Lines 308–311: Please quantify the increase in cloud water content discussed.

**Response:**

We appreciate this important suggestion and have quantified the increase in cloud water content in the revised manuscript (Lines 347–351), as shown below.

➢ Lines 347–351: "At Day −4, the anomalous liquid cloud fraction increased significantly (purple curve in Fig. 4f). The most intense increase prior to heavy rainfall in HA–SP events is observed in the ice water path (orange curve in Fig. 4f), which rises by 22.2–26.8 g m⁻² during Days −3 to −1. This magnitude of increase

is comparable to the cloud water path enhancements under pollution reported in Zhou et al. (2020)."

14. Line 314: The term "modest and insignificant" change in ice cloud fraction should be qualified—how small is the change?

**Response:**

Thank you for this question. In our analysis, differences exceeding the 90% significance level are indicated by gray asterisks in Fig. 4. At Day –3, the ice cloud fraction exhibits a modest increase of ~0.01, which is statistically insignificant. To clarify, we now provide the quantitative value in the revised manuscript (Lines 354–357), as shown below.

➢ Lines 354–357: "Consequently, the ice water path increases markedly (orange curve and gray asterisk in Fig. 4f), whereas the ice cloud fraction shows a modest enhancement of ~0.01 at Day –3. The statistically insignificant increase in ice cloud fraction may reflect a limited reduction in the liquid cloud fraction and persistent supercooled droplets."

15. Line 317: Are uncertainties or confidence intervals available for the vertical profiles shown in Fig. 4?

**Response:**

We appreciate this thoughtful comment. To clarify, the variables shown in Fig. 4 are column-integrated (e.g., ice water path, $Q_2$, radiative budget terms) or bulk properties (e.g., cloud top pressure, cloud fraction), rather than vertical profiles. The figure presents their composite temporal evolution.

Regarding uncertainties, we employed the bootstrap method (Mudelsee et al., 2014) to evaluate statistical significance, as introduced in Section 2.2 (Lines 145–150). In this procedure, paired resampled datasets are generated through random sampling with replacement, metrics are recomputed 1,000 times, and the 90% confidence interval is obtained from the 5th and 95th percentiles. In Fig. 4, only terms with statistically

significant differences at the 90% confidence level are shown, with significant periods marked by gray asterisks (Lines 395–397).

16. Line 357: The latent heating of >30 W m⁻² is substantial. How does this compare with known values for tropical or monsoon convection in prior studies?

**Response:**

Thank you for this good question. The latent heating in Fig. 4i is derived from the moisture budget ($Q_2$, units: kg kg⁻¹ s⁻¹) by vertically integrating from 1000–100 hPa and converting to flux units (~30–40 W m⁻²). To validate the magnitude, we compared our $Q_2$ with the GPM DPR Spectral Latent Heating Profiles product (Shige et al., 2004, 2007, 2008), which shows vertical heating rates of about −1 to 2 K d⁻¹ (Fig. B8). In the revised manuscript (Fig. 5c), we also present $Q_2$ in units of K d⁻¹, showing magnitudes of about −1 to 1 K d⁻¹.

These values are comparable to previous studies: $Q_2$ can reach ~60 W m⁻² in the tropics at intraseasonal timescales (Fu et al., 2020), and ~50 W m⁻² during monsoonal extreme rainfall events (Chyi et al., 2023). Thus, the ~30 W m⁻² obtained here is within the expected range for tropical/monsoon convection. We note, however, that $Q_2$ may slightly differ from actual latent heating because subgrid-scale transport processes can be misinterpreted as latent heating (Yanai et al., 1973). In this study, $Q_2$ is used as a proxy indicator to facilitate comparison with other moisture budget and radiative flux terms.

To prevent misunderstanding, we clarified the definition of $Q_2$ in the revised manuscript (Lines 127–128) and also included comparisons with previous studies (Lines 407–409), as reproduced below.

➤ Lines 127–128: "$Q_2$ represents the latent heating due to condensation/evaporation processes and subgrid-scale moisture flux convergences (Yanai et al., 1973)."

➤ Lines 407–409: "The latent heating magnitude (~40 W m⁻²) is also comparable to values reported in previous studies of tropical intraseasonal convection (~60 W m⁻²; Fu et al., 2020) and monsoonal extreme precipitation (~50 W m⁻²; Chyi et al., 2023)."

[Figure]

Figure B8. Vertical profile of mean latent heating rate over South China during the summer period (May–September) of 2014–2021 derived from (a) GPM_3HSLH_DAY. Panels (b) and (c) show similar profiles of $Q_2$ based on (b) ERA5 and (c) MERRA-2. Units: K d$^{-1}$.

17. Lines 356–360: The conclusion that microphysical effects dominate is based on indirect evidence. Is latent heating (green curve) being used as a proxy for microphysical contributions? If so, this should be explicitly stated. How much greater is latent heating compared to radiative components in numerical terms?

**Response:**

We thank the reviewer for this excellent comment. Latent heating is indeed used here as an indicator of aerosol–cloud microphysical contributions, since it directly results from phase transitions in cloud microphysics. This approach has been widely adopted in previous studies of aerosol–cloud–precipitation interactions (e.g., Xiao et al., 2022; Zhu et al., 2024; Fan et al., 2025). In our analysis, the temporal evolution of aerosol-induced latent heating anomalies (green curve in Fig. 4i) also aligns with ice water path anomalies (orange curve in Fig. 4f), supporting its use as a proxy for microphysical contributions.

Quantitatively, the column latent heating difference exceeds 40 W m$^{-2}$ between HA–SP and LA–SP events, which is roughly seven times larger than the aerosol-induced direct radiative effects and longwave cloud-radiative effects (2–5.5 W m$^{-2}$). In WRF-Chem simulations, the vertical-mean (1000–100 hPa) latent heating difference between CTRL and CLEAN experiments is ~0.8 K d$^{-1}$, about four times greater than the corresponding longwave radiative heating (~0.2 K d$^{-1}$). These consistent

observational and modeling results demonstrate that aerosol-induced microphysical effects dominate over radiative effects.

In the revised manuscript, we explicitly clarified the use of latent heating as a microphysical indicator (Lines 398–399) and added quantitative comparisons with radiative heating (Lines 399–404 and 521–525).

➤ Lines 398–399: "Latent heating, a direct product of phase transition processes, serves as an indicator of aerosol effects on cloud microphysical properties (Zhu et al., 2024; Fan et al., 2025)."

➤ Lines 399–404: "Both the aerosol-induced direct radiative effects (light blue curve in Fig. 4i) and longwave cloud-radiative effects (magenta curve in Fig. 4i), with magnitudes of approximately 2–5.5 W m⁻², are significantly smaller than atmospheric latent heating associated with moisture processes, which exceeds 40 W m⁻² (green curve in Fig. 4i). Quantitatively, aerosol-induced latent heating is approximately seven times greater than aerosol-induced longwave radiative heating."

➤ Lines 521–525: "The reduction in latent heating is much larger than that in radiative heating, with values of ~0.8 K d⁻¹ for vertical-mean (1000–100 hPa) latent heating compared to ~0.2 K d⁻¹ for column longwave heating in the CLEAN experiment relative to CTRL. These modeling results align with observations, showing that aerosol-induced latent heating rates are stronger than aerosol-induced radiative heating rates (Figs. 9a, d)."

18. Lines 393–404 (Fig. 6): The model appears to overestimate precipitation and underestimate AOD. Could the magnitude of these biases be quantified? Could model resolution explain the discrepancies, and how might this affect attribution conclusions?

**Response:**

Thank you for raising these important questions. We have now quantified the biases in Fig. 6 and examined how model resolution affects them.

(1) Magnitude of simulated biases: Over the key region, the CTRL simulation shifts the precipitation frequency distribution toward higher values and yields a mean precipitation of ~10.2 mm d⁻¹ compared with ~7.7 mm d⁻¹ in CPC, corresponding to an

overestimation of about 32%. For aerosols, the CTRL experiment underestimates the mean AOD by ~35%.

These quantitative results have been incorporated into the revised manuscript (Lines 445–448 and 450–452), as shown below.

➢ Lines 445–448: "The precipitation distribution from the CTRL simulation (black bars in Fig. 6a) generally resembles the observed distribution (pink bars in Fig. 6a). However, CTRL yields a mean precipitation of ~10.2 mm d⁻¹ compared with ~7.7 mm d⁻¹ in CPC over the key region, corresponding to an overestimation of about 32%."

➢ Lines 450–452: "Additionally, the CTRL simulation captures the overall pattern of the observed AOD distribution (Fig. 6b), though it underestimates the mean AOD by ~35%."

(2) Effect of model resolution: As motivated by your earlier comment (#5), we conducted nested simulations with an inner 4 km high-resolution domain (Fig. B5a). Although the higher-resolution run reduced the amplitude bias in precipitation (Fig. B5b), it introduced larger temporal discrepancies and a more substantial underestimation of AOD magnitude (Fig. B5c). Thus, increasing resolution does not necessarily improve overall fidelity. Crucially, however, the attribution conclusions are unaffected: the relative contributions of aerosol-induced latent heating versus longwave heating remain at a ratio of ~4:1 (Fig. B6). The discussion of resolution effects on model performance and conclusions has been added to the revised manuscript (Lines 528–534).

➢ Lines 528–534: "These nested simulations consistently demonstrate that aerosols enhance rainfall intensity (Figs. S6b–c) and that aerosol–cloud microphysical effects remain dominant (Fig. S7). Importantly, the ratio of aerosol-induced latent heating to longwave radiative heating remains close to 4:1 (Figs. S7e–f). However, larger biases in AOD and precipitation were evident in the nested simulation (Figs. S6b–c), likely due to error transmission from the parent grid and accumulated uncertainties introduced at finer resolution during long-term integrations (Baklanov et al., 2014; Wang et al., 2016)."

19. Fig. 7: Despite broad agreement between model and observations, timing mismatches are evident. Could the authors discuss the sensitivity of their results to these temporal offsets?

**Response:**

Indeed, we acknowledge that the ensemble-mean simulations show a ~1–2 day offset in the timing of precipitation anomalies relative to CPC observations. However, our conclusions are not sensitive to these temporal mismatches for two reasons. First, our analyses are based on composites aligned to Day 0 (the precipitation peak), which minimizes the potential influence of small phase shifts on the diagnosed relationships. Second, the key physical signal is that positive aerosol anomalies precede enhanced precipitation by several days, and this feature is consistently reproduced across observations and simulations, regardless of a slight shift in peak timing. Hence, these temporal offsets therefore do not alter our main conclusion that aerosol–cloud microphysical effects dominate quasi-biweekly precipitation variability. For clarity, we have added discussion of this point in the revised manuscript (Lines 466–470), as shown below.

➢ Lines 466–470: "Since the analyses are based on composites aligned to the precipitation peak (Day 0), the results are not sensitive to small timing offsets of 1–2 days between model and observations. The robust feature is that positive aerosol anomalies consistently precede enhanced quasi-biweekly precipitation (black and blue curves in Figs. 7a–b), supporting our main conclusions and the reliability of the model simulations employed in this study."

20. Line 414: What is the magnitude and direction of the rainfall bias noted?

**Response:**

We appreciate this suggestion and have clarified the magnitude and direction of the rainfall bias in the revised manuscript (Lines 463–466).

➢ Lines 463–466: "Quantitatively, the simulated precipitation anomalies are overestimated by about 6.5 mm d$^{-1}$ relative to CPC observations during the heavy

rainfall case on 21–23 July 2015 (blue curve and shading in Fig. 7a), and the model fails to reproduce the observed precipitation peak on 19 July."

21. Lines 452–465: Please quantify the differences between high-AOD and low-AOD regimes to support the interpretation.

**Response:**

We appreciate this suggestion and have added quantitative comparisons between high-AOD (CTRL) and low-AOD (CLEAN) regimes in the revised manuscript (Lines 510–520).

➢ Lines 510–520: "Compared with CLEAN experiment, longwave radiative heating exhibits negligible changes below 925 hPa, but increases by 0.1–0.3 K d⁻¹ between 850–500 hPa and above 400 hPa in the CTRL experiment (Fig. 9d), consistent with enhanced cloud fractions (Fig. 8a). To quantify cloud contributions to atmospheric longwave heating, we calculate the longwave cloud-radiative effect using all-sky and clear-sky radiative fluxes based on Eq. (2) (gray bars in Fig. 9e). The development of deep convection triggered by enhanced aerosol emissions reduces OLR by ~10 W m⁻² at the TOA, increases atmospheric longwave radiation by ~9 W m⁻², and rises downward longwave radiation by ~1 W m⁻² at the surface. For shortwave radiation, the increased cloud cover reflects more shortwave radiation back to the TOA (gray bars in Fig. 9f). Additionally, the dominance of scattering aerosols during this rainfall event, as indicated by the high single-scatter albedo, contributes to a small shortwave radiation flux of ~2 W m⁻² through atmosphere (blue bars in Fig. 9f)."

22. Line 465: Sulfate dominance is mentioned—was this confirmed through emission data or WRF-Chem's chemical output?

**Response:**

We apologize for the misleading phrasing. Our intention was not to suggest that sulfate aerosols were the dominant composition, but rather that scattering aerosols dominated this event, with sulfate as one component. As noted in our response to

comment #6, the aerosols over South China in July 2015 are primarily of the scattering type. To ensure precision, we have rephrased the relevant statements in the revised manuscript (Lines 518–520).

➢ Lines 518–520: "Additionally, the dominance of scattering aerosols during this rainfall event, as indicated by the high single-scatter albedo, contributes to a small shortwave radiation flux of ~2 W m$^{-2}$ through atmosphere (blue bars in Fig. 9f)."

23. Lines 499–509: This paragraph would benefit from numerical estimates to support its conclusions.

**Response:**

We appreciate this helpful suggestion and have revised the paragraph by incorporating numerical estimates in the revised manuscript (Lines 565–568 and 573–575).

➢ Lines 565–568: "Under CLEAN conditions, where anthropogenic aerosol concentrations are reduced by 90%, precipitation anomalies decease by ~7 mm d$^{-1}$ due to suppressed cold-phase and mixed-phase processes during the heavy rainfall case on 21–23 July 2015 over the key region."

➢ Lines 573–575: "Quantitatively, the column latent heating rate is reduced by ~0.8 K d$^{-1}$ when anthropogenic aerosols are removed, approximately four times greater than the reduction in column longwave radiative heating (~0.2 K d$^{-1}$)."

**References**

Baklanov, A., Schlünzen, K., Suppan, P., Baldasano, J., Brunner, D., Aksoyoglu, S., Carmichael, G., Douros, J., Flemming, J., Forkel, R., Galmarini, S., Gauss, M., Grell, G., Hirtl, M., Joffre, S., Jorba, O., Kaas, E., Kaasik, M., Kallos, G., Kong, X., Korsholm, U., Kurganskiy, A., Kushta, J., Lohmann, U., Mahura, A., Manders-Groot, A., Maurizi, A., Moussiopoulos, N., Rao, S. T., Savage, N., Seigneur, C., Sokhi, R. S., Solazzo, E., Solomos, S., Sørensen, B., Tsegas, G., Vignati, E., Vogel, B., and Zhang, Y.: Online coupled regional meteorology chemistry models in Europe: current status and prospects, Atmos. Chem. Phys., 14, 317-398, https://doi.org/10.5194/acp-14-317-2014, 2014.

Chanchal, K. and Singh, K. S.: Role of high-resolution modeling system in prediction of heavy rainfall events over Tamil Nadu and Kerala on different global/regional datasets, Model. Earth Syst. Environ., 10, 3827-3843, https://doi.org/10.1007/s40808-024-01979-4, 2024.

Chyi, D., Wang, X., Yu, X., and Zhang, J.: Synoptic-Scale Analysis on Development and Maintenance of the 19–21 July 2021 Extreme Heavy Rainfall in Henan, Central China, J Meteorol Res, 37, 174-191, https://doi.org/10.1007/s13351-023-2914-z, 2023.

Dong, Y., Li, J., Zhang, Z., Zhang, C., and Li, Q.: Aerosol single-scattering albedo derived by merging OMI/POLDER satellite products and AERONET ground observations, Earth Syst. Sci. Data, 17, 3873-3892, https://doi.org/10.5194/essd-17-3873-2025, 2025.

Fan, J., Zhang, Y., Li, Z., Yan, H., Prabhakaran, T., Rosenfeld, D., and Khain, A.: Unveiling Aerosol Impacts on Deep Convective Clouds: Scientific Concept, Modeling, Observational Analysis, and Future Direction, J. Geophys. Res.-Atmos., 130, e2024JD041931, https://doi.org/10.1029/2024JD041931, 2025.

Fu, H., Wang, Y., Xie, Y., Luo, C., Shang, S., He, Z., and Wei, G.: Super Typhoons Simulation: A Comparison of WRF and Empirical Parameterized Models for High Wind Speeds, Appl. Sci., 15, 776, https://doi.org/10.3390/app15020776, 2025.

Fu, Z., Hsu, P.-C., and Liu, F.: Factors Regulating the Multidecadal Changes in MJO Amplitude over the Twentieth Century, J. Climate, 33, 9513-9529, https://doi.org/10.1175/JCLI-D-20-0111.1, 2020.

Hersbach, H., Bell, B., Berrisford, P., Hirahara, S., Horányi, A., Muñoz-Sabater, J., Nicolas, J., Peubey, C., Radu, R., Schepers, D., Simmons, A., Soci, C., Abdalla, S., Abellan, X., Balsamo, G., Bechtold, P., Biavati, G., Bidlot, J., Bonavita, M., De Chiara, G., Dahlgren, P., Dee, D., Diamantakis, M., Dragani, R., Flemming, J., Forbes, R., Fuentes, M., Geer, A., Haimberger, L., Healy, S., Hogan, R. J., Hólm, E., Janisková, M., Keeley, S., Laloyaux, P., Lopez, P., Lupu, C., Radnoti, G., de Rosnay, P., Rozum, I., Vamborg, F., Villaume, S., and Thépaut, J.-N.: The ERA5 global reanalysis, Q. J. Roy. Meteor. Soc., 146, 1999-2049, https://doi.org/10.1002/qj.3803, 2020.

Lee, J., Kim, J., Song, C. H., Kim, S. B., Chun, Y., Sohn, B. J., and Holben, B. N.: Characteristics of aerosol types from AERONET sunphotometer measurements, Atmos. Environ., 44, 3110-3117, https://doi.org/10.1016/j.atmosenv.2010.05.035, 2010.

Liu, Z., Ming, Y., Zhao, C., Lau, N. C., Guo, J., Bollasina, M., and Yim, S. H. L.: Contribution of local and remote anthropogenic aerosols to a record-breaking torrential rainfall event in Guangdong Province, China, Atmos. Chem. Phys., 20, 223-241, https://doi.org/10.5194/acp-20-223-2020, 2020.

Mudelsee, M.: Bootstrap confidence intervals, in: Climate Time Series Analysis, Atmospheric and Oceanographic Sciences Library (Vol. 51), Springer, Cham, Germany, 61-104, https://doi.org/10.1007/978-3-319-04450-7, 2014.

Parra, R.: Effect of Global Atmospheric Datasets in Modeling Meteorology and Air Quality in the Andean Region of Ecuador, Aerosol Air Qual. Res., 22, 210292, https://doi.org/10.4209/aaqr.210292, 2022.

Shige, S., Takayabu, Y. N., Tao, W.-K., and Johnson, D. E.: Spectral Retrieval of Latent Heating Profiles from TRMM PR Data. Part I: Development of a Model-Based Algorithm, J. Appl. Meteor. Climatol., 43, 1095-1113, https://doi.org/10.1175/1520-0450(2004)043<1095:SROLHP>2.0.CO;2, 2004.

Shige, S., Takayabu, Y. N., Tao, W.-K., and Shie, C.-L.: Spectral Retrieval of Latent Heating Profiles from TRMM PR Data. Part II: Algorithm Improvement and Heating Estimates over Tropical Ocean Regions, J. Appl. Meteor. Climatol., 46, 1098-1124, https://doi.org/10.1175/JAM2510.1, 2007.

Shige, S., Takayabu, Y. N., and Tao, W.-K.: Spectral Retrieval of Latent Heating Profiles from TRMM PR Data. Part III: Estimating Apparent Moisture Sink Profiles over Tropical Oceans, J. Appl. Meteor. Climatol., 47, 620-640, https://doi.org/10.1175/2007JAMC1738.1, 2008.

Sun, Y. and Zhao, C.: Distinct impacts on precipitation by aerosol radiative effect over three different megacity regions of eastern China, Atmos. Chem. Phys., 21, 16555-16574, https://doi.org/10.5194/acp-21-16555-2021, 2021.

Wang, L., Zhang, Y., Wang, K., Zheng, B., Zhang, Q., and Wei, W.: Application of Weather Research and Forecasting Model with Chemistry (WRF/Chem) over northern China: Sensitivity study, comparative evaluation, and policy implications, Atmos. Environ., 124, 337-350, https://doi.org/10.1016/j.atmosenv.2014.12.052, 2016.

Wei, J., Li, Z., Cribb, M., Huang, W., Xue, W., Sun, L., Guo, J., Peng, Y., Li, J., Lyapustin, A., Liu, L., Wu, H., and Song, Y.: Improved 1 km resolution PM2.5 estimates across China using enhanced space–time extremely randomized trees, Atmos. Chem. Phys., 20, 3273-3289, https://doi.org/10.5194/acp-20-3273-2020, 2020.

Wei, J., Li, Z., Lyapustin, A., Sun, L., Peng, Y., Xue, W., Su, T., and Cribb, M.: Reconstructing 1-km-resolution high-quality PM2.5 data records from 2000 to 2018 in China: spatiotemporal variations and policy implications, Remote Sens. Environ, 252, 112136, https://doi.org/10.1016/j.rse.2020.112136, 2021.

Wilcox, E. M.: Direct and semi-direct radiative forcing of smoke aerosols over clouds, Atmos. Chem. Phys., 12, 139-149, https://doi.org/10.5194/acp-12-139-2012, 2012.

Xiao, Z., Zhu, S., Miao, Y., Yu, Y., and Che, H.: On the relationship between convective precipitation and aerosol pollution in North China Plain during autumn and winter, Atmos. Res., 271, 106120, https://doi.org/10.1016/j.atmosres.2022.106120, 2022.

Yanai, M., Esbensen, S., and Chu, J.-H.: Determination of Bulk Properties of Tropical Cloud Clusters from Large-Scale Heat and Moisture Budgets, J. Atmos. Sci., 30, 611-627, https://doi.org/10.1175/1520-0469(1973)030<0611:DOBPOT>2.0.CO;2, 1973.

Zhang, Y., Gao, Y., Guo, L., and Zhang, M.: Numerical analysis of aerosol direct and indirect effects on an extreme rainfall event over Beijing in July 2016, Atmos. Res., 264, 105871, https://doi.org/10.1016/j.atmosres.2021.105871, 2021.

Zhao, J., Ma, X., Wu, S., and Sha, T.: Dust emission and transport in Northwest China: WRF-Chem simulation and comparisons with multi-sensor observations, Atmos. Res., 241, 104978, https://doi.org/10.1016/j.atmosres.2020.104978, 2020.

Zhou, S., Yang, J., Wang, W. C., Zhao, C., Gong, D., and Shi, P.: An observational study of the effects of aerosols on diurnal variation of heavy rainfall and associated clouds over Beijing–Tianjin–Hebei, Atmos. Chem. Phys., 20, 5211-5229, https://doi.org/10.5194/acp-20-5211-2020, 2020.

Zhu, A., Xu, H., Deng, J., Ma, J., and Hua, S.: Instant and delayed effects of March biomass burning aerosols over the Indochina Peninsula, Atmos. Chem. Phys., 22, 15425-15447, https://doi.org/10.5194/acp-22-15425-2022, 2022.

Zhu, H., Yang, S., Zhao, H., Wang, Y., and Li, R.: Complex interplay of sulfate aerosols and meteorology conditions on precipitation and latent heat vertical structure, npj Clim. Atmos. Sci., 7, 191, https://doi.org/10.1038/s41612-024-00743-w, 2024.

---

## Referee Report (RR1)

**General Comments:**

This manuscript presents a thorough and comprehensive investigation of aerosol effects on 8–30-day rainfall anomalies, integrating observational datasets and WRF-Chem simulations. The study addresses a timely and relevant topic with significant implications for sub seasonal precipitation prediction.

All comments and suggestions provided in my previous review have been thoroughly addressed in the revised manuscript. The authors have satisfactorily clarified methodological assumptions, improved quantitative analyses, and enhanced the overall clarity and scientific rigour of the study. I am satisfied with the revisions made and agree with the publication of the manuscript in its current form.